# A top-down assessment using OMI NO₂ suggests an underestimate in the NOx emissions inventory in Seoul, South Korea during KORUS-AQ

Daniel L. Goldberg[*,1,2], Pablo E. Saide[3], Lok N. Lamsal[4,5], Benjamin de Foy[6], Zifeng Lu[1,2], Jung-Hun Woo[7], Younha Kim[7], Jinseok Kim[7], Meng Gao[8], Gregory Carmichael[9], and David G. Streets[1,2]

[1]Energy Systems Division, Argonne National Laboratory, Argonne, IL 60439 USA
[2]Consortium for Advanced Science and Engineering, University of Chicago, Chicago, IL 60637, USA
[3]Department of Atmospheric and Oceanic Sciences, Institute of the Environment and Sustainability, University of California – Los Angeles, Los Angeles, CA 90095, USA
[4]Goddard Earth Sciences Technology and Research, Universities Space Research Association, Columbia, MD 21046, USA
[5]NASA Goddard Space Flight Center, Code 614, Greenbelt, MD 20771, USA
[6]Department of Earth and Atmospheric Sciences, Saint Louis University, St. Louis, MO 63108, USA
[7]Konkuk University, 05029 Seoul, South Korea
[8]School of Engineering and Applied Sciences, Harvard University, Cambridge, MA 02138, USA
[9]Department of Chemical and Biochemical Engineering, University of Iowa, Iowa City, IA 52242, USA
*Correspondence to*: Daniel L. Goldberg (dgoldberg@anl.gov)

**Abstract.** In this work, we investigate the $NO_x$ emissions inventory in Seoul, South Korea using a regional Ozone Monitoring Instrument (OMI) $NO_2$ product derived from the standard NASA product. We first develop a regional OMI $NO_2$ product by re-calculating the air mass factors using a high-resolution ($4 \times 4$ km$^2$) WRF-Chem model simulation, which better captures the $NO_2$ profile shapes in urban regions. We then apply a model-derived spatial averaging kernel to further downscale the retrieval and account for the sub-pixel variability. These two modifications yield OMI $NO_2$ values in the regional product that are 1.37 larger in the Seoul metropolitan region and >2 times larger near substantial point sources. These two modifications also yield an OMI $NO_2$ product that is in better agreement with the Pandora $NO_2$ spectrometer measurements acquired during the Korea U.S.-Air Quality (KORUS-AQ) field campaign. $NO_x$ emissions are then derived for the Seoul metropolitan area during the KORUS-AQ field campaign using a top-down approach with the standard and regional NASA OMI $NO_2$ products. We first apply the top-down approach to a model simulation to ensure that the method is appropriate: the WRF-Chem simulation utilizing the bottom-up emission inventory yields a $NO_x$ emission rate of $227 \pm 94$ kton/yr, while the bottom-up inventory itself yields a $NO_x$ emission rate of 198 kton/yr. Using the top-down approach on the regional OMI $NO_2$ product, we derive the $NO_x$ emissions rate from Seoul to be $484 \pm 201$ kton/yr, and a $353 \pm 146$ kton/yr $NO_x$ emissions rate using the standard NASA OMI $NO_2$ product. This suggests an underestimate of 53% and 36% using the regional and standard NASA OMI $NO_2$ products respectively. To supplement this finding, we compare the $NO_2$ and $NO_y$ simulated by WRF-Chem to observations of the same quantity acquired by aircraft and find a model underestimate. When $NO_x$ emissions in the WRF-Chem model are increased by a factor of 2.13 in the Seoul metropolitan area, there is better agreement with KORUS-AQ aircraft observations and the re-calculated OMI $NO_2$ tropospheric columns. Finally, we show that by using a WRF-Chem simulation with an updated emissions inventory to re-calculate the AMF, there are small differences (~8%) in OMI $NO_2$ compared to using the original WRF-Chem simulation to derive the AMF. This suggests that changes in model resolution have a larger effect on the AMF calculation than modifications to the Korean

emissions inventory. Although the current work is focused on South Korea using OMI, the methodology developed in this work can be applied to other world regions using TROPOMI and future satellite datasets (e.g., GEMS and TEMPO) to produce high-quality region-specific top-down $NO_x$ emission estimates.

# 1 Introduction

Nitrogen oxides ($NO_x \equiv NO + NO_2$) are a group of reactive trace gases that are toxic to human health and can transform in the atmosphere into other noxious chemical species. In the presence of abundant volatile organic compounds and strong sunlight, NOx can participate in a series of chemical reactions to accelerate the production of $O_3$, another toxic air pollutant with a longer atmospheric lifetime. $NO_x$ can also transform into particulate nitrate, a component of fine particulate matter ($PM_{2.5}$), an additional health hazard. When fully oxidized in the atmosphere, $NO_x$ transforms into $HNO_3$, a key contributor to acid rain. There are some biogenic emissions of $NO_x$ (e.g., lightning), but the majority of the $NO_x$ emissions are from anthropogenic sources (van Vuuren et al., 2011).

There is a rich legacy of $NO_2$ measurements by remote sensing instruments (Burrows et al., 1999). One of these instruments is the Dutch-Finnish Ozone Monitoring Instrument (OMI), which measures the absorption of solar backscatter in the UV-visible spectral range. $NO_2$ can be observed from space because it has strong absorption features within the 400 – 465 nm wavelength region (Vandaele et al., 1998). By comparing observed spectra with a reference spectrum, the amount of $NO_2$ in the atmosphere between the instrument in low-earth orbit and the surface can be derived; this technique is called differential optical absorption spectroscopy (DOAS) (Platt, 1994).

Tropospheric $NO_2$ column contents from OMI have been used to estimate $NO_x$ emissions from various areas around the globe (Streets et al., 2013; Miyazaki et al., 2017) including North America (Boersma et al., 2008; Lu et al., 2015), Asia (Zhang et al., 2008; Han et al, 2015; Kuhlmann et al., 2015; Liu et al., 2017), the Middle East (Beirle et al., 2011), and Europe (Huijnen et al., 2010; Curier et al., 2014). It has also been used to produce and validate $NO_x$ emission estimates from sectors such as soil (Hudman et al., 2010; Vinken et al., 2014a; Rasool et al., 2016), lightning (Allen et al., 2012; Liaskos et al., 2015; Pickering et al., 2016; Nault et al., 2017), power plants (de Foy et al., 2015), aircraft (Pujadas et al., 2011), marine vessels (Vinken et al., 2014b; Boersma et al., 2015), and urban centers (Lu et al., 2015; Canty et al., 2015; Souri et al., 2016).

With a pixel resolution varying from $13 \times 24$ km$^2$ to $26 \times 128$ km$^2$, the OMI sensor was developed for global to regional scale studies rather than for individual urban areas. Even at the highest spatial resolution of $13 \times 24$ km$^2$, the sensor has difficulty observing the fine structure of $NO_2$ plumes at or near the surface (e.g., highways, power plants, factories, etc.) (Chen et al., 2009; Ma et al., 2013; Flynn et al., 2014), which are often less than 10 km in width (Heue et al., 2008). This can lead to a spatial averaging of pollution (Hilboll et al., 2013). A temporary remedy, until higher spatial resolution satellite instruments are operational, is to use a regional air quality simulation to estimate the sub-pixel variability of OMI pixels. Kim et al. (2016) utilize the spatial variability in a regional air quality model to spatially downscale OMI $NO_2$ measurements using a spatial averaging kernel. The spatial averaging kernel technique has shown to increase the OMI $NO_2$ signal within urban areas, which is in better agreement with observations in these regions (Goldberg et al., 2017).

Furthermore, the air mass factor and surface reflectance used in obtaining the global OMI $NO_2$ retrievals are at a coarse spatial resolution (Lorente et al. 2017; Kleipool et al., 2008). While appropriate for a global operational

retrieval, this is known to cause an underestimate in the OMI $NO_2$ signal in urban regions (Russell et al., 2011). The air mass factors in operational OMI $NO_2$ are calculated using $NO_2$ profile shapes that are provided at a $1.25° \times 1°$ spatial resolution in the NASA product (Krotkov et al. 2017) and $2° \times 3°$ spatial resolution in the DOMINO product (Boersma et al., 2011). Developers of the NASA product provide scattering weights and additional auxiliary information so that users can develop their own tropospheric vertical column product a posteriori (Lamsal et al. 2015). Several users have re-calculated the air mass factor using a regional air quality model (Russell et al., 2011; Kuhlmann et al, 2015; Lin et al., 2015; Goldberg et al., 2017), which can better capture the $NO_2$ profile shapes in urban regions. Other techniques to improve the air mass factor involve correcting for the surface pressure in mountainous terrain (Zhou et al., 2009) and accounting for small-scale heterogeneities in surface reflectance (Zhou et al., 2010; Vasilkov et al., 2017). These a posteriori products have better agreement with ground-based spectrometers measuring tropospheric vertical column contents (Goldberg et al., 2017). When available, observations from aircraft can constrain the $NO_2$ profile shapes used in the air mass factor calculation (Goldberg et al., 2017).

In this paper, we apply both techniques (the spatial averaging kernel and an air mass factor adjustment) to develop a regional OMI $NO_2$ product for South Korea. We then use the regional product with only the air mass factor adjustment to derive $NO_x$ emission estimates for the Seoul metropolitan area using an exponentially modified Gaussian (EMG) function (Beirle et al., 2011; Valin et al., 2013; de Foy et al., 2014; Lu et al., 2015); the methodology is described in-depth in Section 2.5.

**2 Methods**

**2.1 OMI NO₂**

OMI has been operational on NASA's Earth Observing System (EOS) Aura satellite since October 2004 (Levelt et al., 2006). The satellite follows a sun-synchronous, low-earth (705 km) orbit with an equator overpass time of approximately 13:45 local time. OMI measures total column amounts in a 2600 km swath divided into 60 unequal area "field-of-views", or pixels. At nadir (center of the swath), pixel size is $13 \times 24$ km$^2$, but at the swath edges, pixels can be as large as $26 \times 128$ km$^2$. In a single orbit, OMI measures approximately 1650 swaths and achieves daily global coverage over 14 – 15 orbits (99 minutes per orbit). Since June 2007, there has been a partial blockage of the detector's full field of view, which has limited the number of valid measurements by blocking consistent rows of data; this is known in the community as the row anomaly (Dobber et al., 2008): http://projects.knmi.nl/omi/research/product/rowanomaly-background.php.

OMI measures radiance data between the instrument's detector and the Earth's surface. Comparison of these measurements with a reference spectrum (i.e., DOAS technique), enables the calculation of the total slant column density (SCD), which represents an integrated trace gas abundance from the sun to the surface and back to the instrument's detector, passing through the atmosphere twice. For tropospheric air quality studies, vertical column density (VCD) data are more useful. This is done by subtracting the stratospheric slant column from the total

(tropospheric + stratospheric) slant column and dividing by the tropospheric air mass factor (AMF), which is defined as the ratio of the SCD to the VCD, as shown in Eq. (1):

$$VCD_{trop} = \frac{SCD_{total} - SCD_{strat}}{AMF_{trop}} \qquad , \text{ where } AMF_{trop} = \frac{SCD_{trop}}{VCD_{trop}} \qquad (1)$$

The tropospheric AMF has been derived to be a function of the optical atmospheric/surface properties (viewing and solar angles, surface reflectivity, cloud radiance fraction, and cloud height) and a priori profile shape (Palmer et al., 2001; Martin et al., 2002) and can be calculated as follows (Lamsal et al., 2014) in Eq. (2):

$$AMF_{trop} = \frac{\sum_{n=surface}^{tropopause} SW_n \times x_n}{\sum_{n=surface}^{tropopause} x_n} \qquad (2)$$

where x is the partial column. The optical atmospheric/surface properties in the NASA retrieval are characterized by the scattering weight and are calculated by a forward radiative transfer model (TOMRAD), which are output as a look-up table. The scattering weights are then adjusted real-time depending on observed viewing angles, surface albedo, cloud radiance fraction, and cloud pressure.

For this study, we follow previous studies (e.g., Palmer et al., 2001, Martin et al., 2002, Boersma et al., 2011, Bucsela et al., 2013) and assume that scattering weights and trace gas profile shapes are independent. The a priori trace gas profile shapes ($x_a$) must be provided by a model simulation. In an operational setting, NASA uses a monthly-averaged and year-specific Global Model Initiative (GMI) global simulation with a spatial resolution of 1.25° lon × 1° lat (~110 km × 110 km in the mid-latitudes) to provide the a priori profile shapes.

We derive tropospheric VCDs using a priori $NO_2$ profile shapes from a regional WRF-Chem simulation. A full description of this methodology can be found in Goldberg et al. (2017); it is also described in brief in section 2.1.1. We filter the Level 2 OMI $NO_2$ data to ensure only valid pixels are used. Daily pixels with solar zenith angles ≥ 80°, cloud radiance fractions ≥ 0.5, or surface albedo ≥ 0.3 are removed as well as the five largest pixels at the swath edges (i.e., pixel numbers 1 – 5 and 56 – 60). Finally, we remove any pixel flagged by NASA including pixels with NaN values, 'XTrackQualityFlags' ≠ 0 or 255 (RA flag), or 'VcdQualityFlags' > 0 and least significant bit ≠ 0 (ground pixel flag).

**2.1.1 OMI-WRF-Chem $NO_2$**

We modify the air mass factor in the OMI $NO_2$ retrieval based on the vertical profiles from a high spatial (4 × 4 $km^2$) resolution WRF-Chem simulation. The vertical profiles are scaled based on a comparison with in situ aircraft observations; this accounts for any consistent biases in the model simulation. For example, if the aircraft observations show that mean $NO_2$ concentrations between 0 - 500 m are low by 50%, then we scale the modeled $NO_2$ in this altitude bin by this same amount. To re-calculate the air mass factor for each OMI pixel, we first compute sub-pixel air mass factors for each WRF-Chem model grid cell, using the same method as outlined in Goldberg et al. (2017). The sub-

pixel air mass factor for each WRF-Chem grid cell is a function of the modelled $NO_2$ profile shape and the scattering weight calculated by a radiative transfer model. We then average all sub-pixel air mass factors within an OMI pixel (usually 10-100) to generate a single tropospheric air mass factor for each individual OMI pixel. This new air mass factor is used to convert the total slant column into a total vertical column using Equation 1. Model outputs were sampled at the local time of OMI overpass. For May 2016, we used daily $NO_2$ profiles and terrain pressures (e.g., (Zhou et al., 2009, Laughner et al., 2016)) to re-calculate the AMF. For other months and years, we used May 2016 monthly mean values of $NO_2$ and tropopause pressures for the a priori profiles, which are used in the calculation of the AMF.

Once the tropospheric vertical column of each OMI pixel was re-calculated, the product was oversampled (de Foy et al., 2009; Russell et al., 2010) for April – June over a 3-year period (2015-2017; 9 months total). During this timeframe, there are approximately 9 valid OMI $NO_2$ pixels per month over any given location on the Korean peninsula. In the top-down emissions derivation, we use all nine-months of OMI data for the analysis.

**2.2 $NO_2$ observations during KORUS-AQ**

We use in situ $NO_2$ observations from the KORUS-AQ field campaign to test the regional satellite product. KORUS-AQ was a joint Korea-US field experiment designed to better understand the trace gas and aerosol composition above the Korean peninsula using aircrafts, ground station networks, and satellites. The campaign took place between May 1, 2016 and June 15, 2016 and measurements were primarily focused in the Seoul Metropolitan Area. In this paper, we utilize data acquired by the ground-based Pandora spectrometer network, the thermally dissociated laser-induced fluorescence $NO_2$ instrument on DC-8 aircraft, and the chemiluminescence $NO_y$ instrument on the DC-8 aircraft ($NO_y$ = $NO + NO_2 + HNO_3 + 2 \times N_2O_5$ + peroxy nitrates + alkyl nitrates + …). KORUS-AQ observations were retrieved from the online data archive: http://www-air.larc.nasa.gov/cgi-bin/ArcView/korusaq. A further description of this field campaign can be found in the KORUS-AQ White Paper (https://espo.nasa.gov/korus-aq/content/KORUS-AQ_Science_Overview_0).

**2.2.1 Pandora $NO_2$ data**

Measurements of total column $NO_2$ from the Pandora instrument (Herman et al., 2009; Herman et al., 2018) are used to evaluate the OMI $NO_2$ satellite products. The Pandora instrument is a stationary, ground-based, sun-tracking spectrometer, which measures direct sunlight in the UV-Visible spectral range (280-525 nm) with a sampling period of 90 seconds. The Pandora spectrometer measures total column $NO_2$ using a DOAS technique similar to OMI. A distinct advantage of the Pandora instrument is that it does not require complex assumptions for converting slant columns into vertical columns, compared to zenith sky measurements (e.g., MAX-DOAS).

Valid OMI $NO_2$ pixels are matched spatially and temporally to Pandora total column $NO_2$ observations. To smooth the data and eliminate brief small-scale plumes that would be undetectable by a satellite, we average the Pandora observations over a two hour period (± one hour of the overpass time) before matching to the OMI $NO_2$ data (Goldberg

et al., 2017).  During May 2016, there were seven Pandora $NO_2$ spectrometers operating during the experiment (five instruments were situated within the Seoul metropolitan area and their locations are shown in Figure 5); this corresponded to fifty instances in which valid Pandora $NO_2$ observations matched valid OMI $NO_2$ column data.

**2.2.2 DC-8 aircraft data**

We compare the model simulation to in situ $NO_2$ data gathered by the UC-Berkeley Cohen group (Thornton et al., 2000; Day et al, 2002) on the DC-8 aircraft.  The instrument quantifies $NO_2$ via laser-induced fluorescence at 585 nm. This instrument does not have the same positive bias as chemiluminescence $NO_2$ detectors, so there is no need to modify $NO_2$ concentrations by applying an empirical equation (e.g., Lamsal et al., 2008).  We also compare the model simulation to chemiluminescence $NO_y$ data gathered by the NCAR Weinheimer group (Ridley et al., 2004)

We utilize one-minute averaged DC-8 data from all fourteen flights during May – June 2016.  A typical flight path included several low-altitude spirals over the Seoul Metropolitan Area and a long-distance transect over the Korean peninsula or the Yellow Sea.  One-minute averaged data is already pre-generated in the data archive.  Hourly output from the model simulation is spatially and temporally matched to the observations.  We then bin the data into different altitude ranges for our comparison.

**2.3 WRF-Chem model simulation**

For the high-resolution OMI $NO_2$ product, we use a regional simulation of the Weather Research & Forecasting (Skamarock et al., 2008) coupled to Chemistry (WRF-Chem) (Grell et al., 2005) in forecast mode prepared for flight planning during the KORUS-AQ field campaign.  The forecast simulations were performed daily and used National Centers for Environmental Prediction Global Forecast System (https://rda.ucar.edu/datasets/ds084.6/) meteorological

initial and boundary conditions from the 06 UTC cycle.  Initial conditions for aerosols and gases were obtained from the previous forecasting cycle, while Copernicus Atmosphere Monitoring Service (Inness et al., 2015) forecasts were used as boundary conditions.  WRF-Chem was configured with two domains, with 20 km and 4 km grid-spacing.  The 20 km domain included the major sources for trans-boundary pollution impacting the Korean peninsula (deserts in China and Mongolia, wild-fires in Siberia and anthropogenic sources from China).  The 4 km domain provided a high-

resolution simulation where detailed local sources could be modeled and where the KORUS-AQ flight tracks were contained.  The inner domain was started 18 hours after the outer domain, and was simulated for 33 hours (00UTC from day 1 to 9 UTC of day 2 of the forecast); output was saved hourly.  The last 24 hours of each inner domain daily forecast over the course of KORUS-AQ were selected to allow spin-up from the outer domain and were used in the analysis presented here.

WRF-Chem was configured with 4 bin MOSAIC aerosols (Zaveri et al., 2008), a reduced hydrocarbon trace gas chemical mechanism (Pfister et al., 2014) including simplified secondary organic aerosol formation (Hodzic and Jimenez, 2011), and with capabilities to assimilate satellite aerosol optical depth both from low-earth orbiting and geostationary satellites (Saide et al., 2013, 2014).

**2.4 Emission Inventory**

The WRF-Chem simulation was driven by emissions developed by Konkuk University. Monthly emissions for South Korea were developed using the projected 2015 Korean national emissions inventory, Clean Air Policy Support System (CAPSS) provided by the National Institute of Environmental Research of Korea and with enhancements by Konkuk University, which primarily include the addition of new power plants. The projected CAPSS 2015 emissions were estimated based on CAPSS 2012 and 3-year growth factors. Since the base year of the inventory is 2012, observed emissions from the post-2013 Large Point Source inventory were not included. Emissions from China and North Korea were taken from the Comprehensive Regional Emissions for Atmospheric Transport Experiments (CREATE) v3.0 emissions inventory. In order to project the year 2010 emissions to 2015, the latest energy statistics from the International Energy Agency (http://www.iea.org/weo2017/) and the China Statistical Yearbook 2016 (http://www.stats.gov.cn/tjsj/ndsj/2016/indexeh.htm) were used to update the growth of fuel activities. In addition, the new emissions control policies in China, which were compiled by the International Institute for Applied Systems Analysis, were applied to consider efficiencies of emissions control (van der A et al., 2017).

Emissions were first processed to the monthly time-scale at a spatial resolution of 3 km in South Korea and 0.1° for the rest of Asia using SMOKE-Asia (Woo et al., 2012). Information from GIS, such as population, road network, and land cover, were applied to generate gridded emissions from the region-based (17 metropolitan and provincial boundaries of South Korea) emissions. The GIS-based population and regional boundary data compiled by the Ministry of Interior and Safety (http://www.mois.go.kr/frt/sub/a05/totStat/), and land cover data compiled by the Ministry of Environment (https://egis.me.go.kr/) were used to generate population and land cover based spatial surrogates. The Road and Railroad network data compiled by The Korea Transport Institute were used to generate spatial surrogates for onroad and nonroad emissions (https://www.koti.re.kr/). The emissions were downscaled temporally from monthly to hourly and spatially re-allocated to 4 km over South Korea and 20 km over the rest of East Asia using the University of Iowa emission pre-processor (EPRES).

Biogenic emissions are included using the on-line Model of Emissions of Gases and Aerosols from Nature (MEGAN) model version 2; there are no $NO_x$ emissions from MEGAN. For this simulation, the lightning $NO_x$ parameterization was turned off. For wildfires we used the Quick Fire Emissions Dataset (QFED2), but there were only isolated, small fires in South Korea during this timeframe.

**2.5 Exponentially Modified Gaussian Fitting Method**

An exponentially modified Gaussian (EMG) function is fit to a collection of $NO_2$ plumes observed from OMI in order to determine the $NO_2$ burden and lifetime from the Seoul metropolitan area. The original methodology, proposed by Beirle et al. (2011), involves the fitting of OMI $NO_2$ line densities to an EMG function. OMI $NO_2$ line densities are the integral of OMI $NO_2$ retrieval perpendicular to the path of the plume; the units are mass per distance. We define integration length scale as the across plume width. The across plume width is dependent on the $NO_2$ plume size and

can vary between 10 km (for small point sources) to 240 km (for large urban areas). Visual inspection of the rotated oversampled OMI $NO_2$ plumes is the best way to determine the spatial extent of the emission sources (Lu et al. 2015).

The EMG model is expressed as Equation (3):

$$OMI\ NO_2\ Line\ Density = \alpha \left[ \frac{1}{x_o} exp \left( \frac{\mu}{x_o} + \frac{\sigma^2}{2x_o^2} - \frac{x}{x_o} \right) \Phi \left( \frac{x-\mu}{\sigma} - \frac{\sigma}{x_o} \right) \right] + \beta \qquad (3)$$

where $\alpha$ is the total number of $NO_2$ molecules observed near the hotspot, excluding the effect of background $NO_2$, $\beta$; $x_o$ is the e-folding distance downwind, representing the length scale of the $NO_2$ decay; $\mu$ is the location of the apparent source relative to the city center; $\sigma$ is the standard deviation of the Gaussian function, representing the Gaussian smoothing length scale; $\Phi$ is the cumulative distribution function. Using the 'curvefit' function in IDL, we determine the five unknown parameters: $\alpha$, $x_o$, $\sigma$, $\mu$, $\beta$ based on the independent (distance; x) and dependent (OMI $NO_2$ line

density) variables.

Using the mean zonal wind speed, $w$, of the $NO_2$ line density domain, the mean effective $NO_2$ lifetime $\tau_{effective}$ and the mean NOx emissions can be calculated from the fitted parameters $x_o$ and $\alpha$. The wind speed and direction are obtained from the ERA-Interim re-analysis project (Dee et al., 2011), instead of the WRF simulation because the WRF simulation is a forecast. We use the averaged wind fields of the bottom eight levels of the re-analysis (i.e., from the

surface to ~500 m). Only days in which the wind speeds are > 3 m/s are included in this analysis, because $NO_2$ decay under this condition is dominated by chemical removal, not variability in the winds (de Foy et al., 2014). The factor of 1.33 is the mean column-averaged $NO_x$ / $NO_2$ ratio in the WRF-Chem model simulation for the Seoul metropolitan area during the mid-afternoon. The $NO_x$ / $NO_2$ ratio is time-dependent, spatially varying and is primarily a function of the localized $j(NO_2)$ and $O_3$ concentration.

$$NO_x\ Emissions = 1.33 \left( \frac{\alpha}{\tau_{effective}} \right),\ \text{where } \tau_{effective} = \frac{x_o}{w} \qquad (4)$$

The $NO_2$ plume concentration is a function of the emission source strength, wind speed, and wind direction. Originally, the method separated all $NO_2$ plumes by wind direction, and fit an EMG function to $NO_2$ in eight wind directions (Beirle et al., 2011; Ialongo et al., 2014; Liu et al., 2016). Newer methodologies rotate the plumes so that all plumes are in the same direction (Valin et al., 2013; de Foy et al., 2014; Lu et al., 2015). This process increases the

signal-to-noise ratio and generates a more robust fit. In this work, we filter OMI $NO_2$ data and rotate the $NO_2$ plumes and as described in Lu et al. (2015), so that all plumes are decaying in the same direction. We rotate the retrieval based on the re-analyzed 0-500 m wind speed direction from the ERA-Interim. In doing so, we develop a re-gridded satellite product in an x-y coordinate system, in which the urban plume is aligned along the x-axis. Following de Foy et al. (2014) and Lu et al. (2015), we only use days in which the ERA-Interim wind speeds are > 3 m/s because there is

more direct plume transport and less plume meandering on days with stronger winds; this yields more robust $NO_x$ emission estimates. We fit an EMG function to the line density as function of the horizontal distance. This yields a single value at each point along the x-direction.

**3 Results**

In this section, we describe the regional high-resolution satellite product and our validation efforts. All OMI NO$_2$ results presented here are vertical column densities. First, we show a continental snapshot of OMI NO$_2$ (OMI-Standard) over East Asia using the standard NASA product. Then, we show a regional NASA OMI NO$_2$ satellite
product (OMI-Regional) using AMFs generated from the WRF-Chem a priori NO$_2$ profiles. We compare the OMI-Regional product with NO$_2$ VCDs from the original WRF-Chem simulation. We evaluate the OMI-Regional product by comparing to KORUS-AQ observations. Finally, we use the OMI-Standard and OMI-Regional products to estimate NO$_x$ emissions from the Seoul metropolitan area.

**3.1 OMI NO$_2$ in East Asia**

Oversampled OMI NO$_2$ for May – September 2015 – 2017 (15 months total) in East Asia and the 4 km WRF-Chem model domain are shown in Figure 1. The OMI NO$_2$ signals in East Asia over major metropolitan areas are 3 to 5 times larger than over similarly sized cities in the US (Krotkov et al., 2016). This is in despite of recent NO$_x$ reductions in China since 2011 (de Foy et al., 2016; Souri et al., 2017; Zheng et al., 2018). OMI has observed a recent decrease in the NO$_2$ burden in the immediate Seoul, South Korea metropolitan area, but an increase in areas just outside the
city center (Duncan et al., 2016). Oversampled values greater than $8 \times 10^{15}$ molecules per cm$^2$ are still consistently seen in East Asia, while they are non-existent in the US during the warm season.

**3.2 Calculation of new OMI tropospheric column NO$_2$**

In Figure 2, we plot the OMI-Standard and OMI-Regional products over South Korea. The top center panel shows a regional product in which only the air mass factor correction is applied (AMF). The bottom center panel shows a
regional product in which the air mass factor correction and spatial averaging kernel are applied (AMF+SK). The regional product yields larger OMI NO$_2$ values throughout the majority of the Korean peninsula. Areas near major cities (e.g. Seoul), power plants, steel mills, and cement kilns have OMI NO$_2$ values that are >1.25 times larger in the regional AMF product and >2 times larger in the regional AMF+SK product. There are two reasons for the larger OMI NO$_2$ signals: the air mass factors in polluted regions are now smaller (Russell et al., 2011; Goldberg et al., 2017)
and the spatial weighting kernel allocates a large portion of the OMI NO$_2$ signal into a smaller region (Kim et al., 2016).

**3.3 OMI-Regional vs. WRF-Chem**

We now compare the OMI-Regional product to tropospheric vertical columns from the WRF-Chem model simulation directly. In Figure 3, we compare the regional satellite product (AMF+SK) to the WRF-Chem simulation over the
Korean peninsula. In most areas, the modeled tropospheric column NO$_2$ is of smaller magnitude than inferred by the satellite. In the area within 40 km of the Seoul city center, modeled tropospheric vertical columns are 44% smaller than observed tropospheric vertical column in the regional AMF+SK product. We posit four reasons as to why the model simulation calculates columns that are consistently smaller. First, WRF-Chem uses a reduced hydrocarbon

gas-phase chemical mechanism. This fast-calculating mechanism implemented in WRF-Chem for regional climate assessments (Pfister et al., 2014) and used during KORUS-AQ for forecasting does not quickly recycle alkyl nitrates back to $NO_2$; this will cause $NO_2$ to be too low. While an underestimate of the chemical conversion to $NO_2$ in WRF-Chem is a contributor to the underestimate, it likely does not account for the entire discrepancy; Canty et al., (2015) suggests that by shortening the lifetime of alkyl nitrates in the chemical mechanism, $NO_2$ will increase by roughly 3% in urban areas and 18% in rural areas. Second, an underestimate in VOC emissions would have an impact on peroxyacyl and alkyl nitrate formation, and should enhance the effective $NO_x$ lifetime (Romer et al., 2016). Third, the temporal allocation of $NO_x$ emissions in this WRF-Chem simulation is such that the early afternoon rate (between 12:00 – 14:00 local time) is approximately equal to 24-hour averaged rate (Figure 4). For comparison, using SMOKE in the eastern US yields an early afternoon emission rate that is 1.35 larger than the 24-hour averaged emission rate. Lastly, the remaining difference will likely be due to an underestimate in the emissions inventory.

### 3.4 Comparing WRF-Chem to Aircraft Measurements

When comparing the model simulation to in situ observations from the UC-Berkeley $NO_2$ instrument aboard the aircraft, we find that $NO_2$ concentrations are substantially larger than the model when spatially and temporally co-located in the immediate Seoul metropolitan area (Figure 5). The comparison isolates the $NO_2$ within the lowermost boundary layer as the primary contributor to the tropospheric column underestimate. When comparing aircraft $NO_2$ to modeled $NO_2$ in other areas of the Korean peninsula, the underestimate is smaller.

When comparing the model simulation of $NO_y$ to observations of the same quantity observed from the aircraft, we find a similarly large underestimate. $NO_y$ observed on the aircraft is roughly a factor of two larger at all altitudes below 2 km. This suggests that errors in $NO_2$ recycling ($NO_2 \leftrightarrow NO_y$) are not the main cause of the $NO_2$ discrepancies seen in the satellite and aircraft comparison. Instead, there must be errors in the $NO_y$ production (i.e., $NO_x$ emission rates are too low) or removal rates (i.e., $NO_y$ deposition rates are too slow).

### 3.5 Comparison of OMI $NO_2$ to Pandora $NO_2$

To quantify the skill of the regional OMI $NO_2$ product, we compare the new total $NO_2$ vertical columns from the satellite product to the same quantities observed by the Pandora instruments. In Figure 6, monthly averaged observations during May 2016 from the Pandora instrument are overlaid onto the monthly average of the three OMI $NO_2$ satellite products. The two regional OMI $NO_2$ products capture the magnitude and spatial variability of monthly averaged $NO_2$ within the metropolitan region better.

We then compare daily Pandora observations to each daily OMI $NO_2$ value spatially and temporally co-located with the Pandora instrument (Figure 6). The Pandora observation is a 2-hour mean centered on the mid-afternoon OMI overpass. The slope of the linear best-fit of the standard product is 0.58, indicating that there is a consistent low bias in the satellite product when the Pandora instrument observes large values. The best-fit slope of the OMI-Regional product with only the air mass factor adjustment (AMF) is 0.76, and the OMI-Regional product with the air mass

factor adjustment and spatial kernel (AMF+SK) is 1.07, indicating that the regional products capture the polluted-to-clean spatial gradients best. The correlation of daily observations to the satellite retrievals does not improve between retrievals (OMI-Standard: $r^2 = 0.57$, OMI-Regional (AMF): $r^2 = 0.57$, and OMI-Regional (AMF+SK): $r^2 = 0.58$). The lack of improvement in the correlation suggests that the forecasted WRF-Chem simulation is unable to capture the daily variability of $NO_2$ plumes better than a longer-term average.

### 3.6 Estimating $NO_x$ emissions from Seoul

To estimate $NO_x$ emissions from the Seoul metropolitan area using a top-down satellite-based approach, we follow the exponentially modified Gaussian (EMG) fitting methodology outlined in Section 2.5. When fit using the EMG method, the photochemical lifetime and OMI $NO_2$ burden can be derived. Using this information, a $NO_x$ emission rate can be inferred.

### 3.6.1. Validating the EMG method using WRF-Chem

The WRF-Chem simulation can serve as a test bed to assess the accuracy of the EMG method, since the bottom-up emissions used for the simulation are known. For this study, we find that for Seoul, an across plume width of 160 km encompasses the entire $NO_2$ downwind plume. Using the $NO_2$ lifetime, $NO_2$ burden, and a 160 km across plume width, we calculate the top-down $NO_x$ emissions rate in the WRF-Chem simulation from the Seoul metropolitan area during the early afternoon (Figure 7). We find the effective $NO_2$ photochemical lifetime to be $3.1 \pm 1.3$ hours and the emissions rate to be $227 \pm 94$ kton/yr $NO_2$ equivalent. Uncertainties of the top-down $NO_x$ emissions are the square root of the sum of the squares of: the $NO_x / NO_2$ ratio (10%), the OMI $NO_2$ vertical columns (25%), the across plume width (10%), and the wind fields (30%) (Lu et al., 2015). Only the latter three terms are used to calculate the uncertainty of the $NO_2$ lifetime (Lu et al., 2015).

The $NO_x$ bottom-up emissions inventory calculated using a 40 km radius from the Seoul city center is 198 kton/yr $NO_2$ equivalent. We use a 40 km radius in lieu of a larger radius because an assumption in EMG method is that the emissions must be clustered around a single point (in this case, the city center). Therefore, the calculated emissions rate from the EMG fit is only measuring the magnitude of the perturbing emission source, and not of smaller sources that are further from the city center. Previous studies (de Foy et al., 2014; de Foy et al., 2015) suggest that the background level calculated by the EMG fit accounts for emissions outside the plume that are more regional and diffuse in nature. The agreement between the top-down (227 kton/yr) and bottom-up (198 kton/yr) approaches demonstrates the accuracy and effectiveness of the EMG method in estimating the emissions rate.

### 3.6.2. Deriving emissions using OMI $NO_2$

We now calculate the top-down $NO_x$ emissions rate from the satellite data from the Seoul metropolitan area during the early afternoon (Figure 8). Here we use the OMI standard product and the OMI $NO_2$ retrieval without the spatial averaging kernel; only the new air mass factor is applied to this retrieval. We do not use the retrieval with the spatial averaging kernel when calculating top-down $NO_x$ emissions because the spatial averaging is strongly dependent on

the wind fields in the WRF-Chem simulation, which are forecasted. Errors in the winds can greatly affect the estimate using this top-down approach (Valin et al., 2013; de Foy et al., 2014).

For the standard product, the effective $NO_2$ photochemical lifetime is $4.2 \pm 1.7$ hours, while in the regional product, the effective lifetime is $3.4 \pm 1.4$ hours. In the standard product, we derive the $NO_x$ emissions rate to be $353 \pm 146$ kton/yr $NO_2$ equivalent, while in the regional product it is $484 \pm 201$ kton/yr $NO_2$ equivalent. Emission estimates using satellite products with coarse resolution air mass factors will yield top-down emission estimates that are lower than reality. In this case, the regional satellite product yields $NO_x$ emission rates that are 37% higher; we would expect similar results from other metropolitan regions. The top-down approach for the model simulation yielded a $NO_x$ emission rate of 227 kton/yr, while the top-down approach using the satellite data yielded a 484 kton/yr $NO_x$ emission rate: a 53% underestimate in the emissions inventory.

It should be noted that the $NO_2$ photochemical lifetime derived here is a fundamentally different quantity than the $NO_2$ lifetime observed by in situ measurements (de Foy et al., 2014; Lu et al., 2015) or derived by model simulations (Lamsal et al., 2010). This is because the lifetime calculation is extremely sensitive to the accuracy of the wind direction (de Foy et al., 2014). Inaccuracies in the wind fields introduce noise that shorten the tail of the fit. As a result, $NO_2$ photochemical lifetimes derived here are considered "effective" photochemical lifetimes and are universally shorter than the tropospheric column $NO_2$ lifetimes derived by model simulations (Lamsal et al., 2010). NOx sources at the outer portions of urban areas will lead to an artificially longer $NO_2$ lifetime. This partially compensates for the bias introduced by the wind direction. The effective photochemical lifetime is also different from the $NO_2$ lifetime derived by in situ measurements of $NO_2$ at the surface or within the boundary layer. In the boundary layer, $NO_2$ is consumed faster yielding lifetimes that are shorter than the lifetimes based on tropospheric columns (Nunnermacker et al., 2007).

**3.7. Model simulation with increased $NO_x$ emissions**

To test whether an increase in the $NO_x$ emission rate is appropriate for the Seoul metropolitan area, we conduct a simulation with $NO_x$ emissions in the Seoul metropolitan area – within a 40 km radius of the city center – increased by a factor of 2.13, and analyze the results for May 2016. The 2.13 increase is representative of the change suggested by the top-down method (OMI-Regional: 484 kton/yr vs. WRF-Chem original: 227 kton/yr). This simulation was performed slightly differently than the original simulation in that it was a continuous month-long simulation and the outer domain was nudged to the reanalysis.

When comparing the new model simulation to in situ observations from the UC-Berkeley $NO_2$ and NCAR $NO_y$ instruments aboard the DC-8 aircraft, we find that $NO_2$ concentrations are a bit high, but $NO_y$ concentrations are in good agreement with WRF-Chem in the boundary layer when spatially and temporally co-located in the immediate Seoul metropolitan area (Figure 9). When comparing the new WRF-Chem simulation to the OMI-Regional product for May 2016 (Figure 10), we find no significant biases in the Seoul metropolitan area. In the area within 40 km of

the Seoul city center, $NO_2$ columns are now only 11% smaller. The better agreement in $NO_2$ and $NO_y$ from a combination of aircraft and satellite data suggests that an increase in $NO_x$ emissions by a factor of 2.13 is appropriate.

Finally, we re-process the air mass factors for May 2016 using the newest WRF-Chem simulation. In Figure 11, we show the OMI-Standard product, the OMI-Regional product with no scaling of the a priori profiles from the original WRF-Chem simulation, the OMI-Regional product with scaling of the original a priori profiles, and the OMI-Regional product with a priori profiles from the new WRF-Chem simulation. While using the new a priori profiles increases the OMI $NO_2$ retrieval further by 8%, this change is much smaller than the 37% increase associated with switching models and model resolution (i.e., Standard vs. Regional product).

## 4. Conclusions and Discussion

In this work, we use a high-resolution ($4 \times 4$ km$^2$) WRF-Chem model simulation to re-calculate satellite $NO_2$ air mass factors over South Korea. We also apply a spatial averaging kernel to better account for the sub-pixel variability that cannot be observed by OMI. The regional OMI $NO_2$ retrieval yields increased tropospheric columns in city centers and near large industrial areas. In the area within 40 km of the Seoul city center, OMI $NO_2$ values are 1.37 larger in the regional product. Areas near large industrial sources have OMI $NO_2$ values that are >2 times larger. The increase in remotely sensed tropospheric vertical column contents in the Seoul metropolitan area is in better agreement with the Pandora $NO_2$ spectrometer measurements acquired during the KORUS-AQ field campaign.

Using the regional OMI $NO_2$ product with only the air mass factor correction applied, we derive the $NO_x$ emissions rate from the Seoul metropolitan area to be $484 \pm 201$ kton/yr, while the standard NASA OMI $NO_2$ product gives an emissions rate of $353 \pm 146$ kton/yr. The WRF-Chem simulation yields a mid-afternoon $NO_x$ emission rate of $227 \pm 94$ kton/yr. This suggests an underestimate in the bottom-up $NO_x$ emissions from Seoul metropolitan area by 53%, when compared to the 484 kton/yr emissions rate from our top-down method. When comparing observed OMI $NO_2$ to the WRF-Chem model simulation, we find similar underestimates of $NO_2$ in the Seoul metropolitan area. The effective photochemical lifetime derived in the Seoul plume is $4.2 \pm 1.7$ hours using the standard OMI $NO_2$ product and $3.4 \pm 1.4$ hours using the regional product. The regional product yields shorter $NO_2$ lifetimes, although it is not a statistically significant difference. Finally, we show that a WRF-Chem simulation with an increase in the $NO_x$ emissions by a factor of 2.13 yields a better comparison with aircraft observations of $NO_2$ and $NO_y$, and is in better agreement with the OMI-Regional $NO_2$ product developed herein.

It should be noted that the Seoul metropolitan area has complex geographical features, which adds further uncertainty to this analysis. The area has large topographical changes over short distances, including many hills (> 500 m) within the metropolitan area. Furthermore, the city is in close proximity to the Yellow Sea, which causes the area to be affected by sea breeze fronts, especially in the springtime, which is our period of focus. The localized mountain and sea breezes may not be fully captured by our $4 \times 4$ km$^2$ WRF-Chem simulation used to derive the OMI-Regional product or the ERA-interim dataset used to calculate top-down $NO_x$ emissions. The effects of these features on local air quality have been documented elsewhere in the literature (Kim and Ghim, 2002; Lee et al., 2008; Ryu et al., 2013).

We do not expect any consistent bias to result from this added uncertainty. Nevertheless, the $4 \times 4$ km$^2$ simulation will capture topography and mesoscale phenomena better than a coarse global model and further supports the benefits of WRF-Chem over a global model to derive $NO_2$ vertical column contents.

We hypothesize that the temporalization of $NO_x$ emissions in the bottom-up emission inventory is a large remaining
5      uncertainty. The satellite-derived emission rates are instantaneous rates at the time of the OMI overpass (~13:45 local time). This is a different quantity than a bottom-up $NO_x$ emission inventory, which is often a daily averaged or monthly averaged emission rate. For this study, we only attempt to derive a mid-afternoon $NO_x$ emission rate. Subsequently, we make sure to compare this to the mid-afternoon $NO_x$ emission rate from WRF-Chem. While bottom-up studies provide estimates of the diurnal variability of $NO_x$ emissions, these are very difficult to confirm
10     from top-down approaches. Due to a consistent mid-afternoon overpass time, OMI or TROPOMI cannot address this issue. Due to boundary layer dynamics, this is also very difficult to constrain from ground-based and aircraft measurements. In the future, observations from a geostationary satellite instruments such as the Geostationary Environment Monitoring Spectrometer (GEMS) and Tropospheric Emissions: Monitoring Pollution (TEMPO), will be helpful in constraining the ratio of the mid-afternoon emissions rate to the 24-hour averaged emission rate.

**Acknowledgments**

This publication was developed using funding from the NASA KORUS-AQ science team and the NASA Atmospheric Composition Modeling and Analysis Program (ACMAP). We would like to thank the NASA Pandora Project Team, including Jay Herman and Bob Swap of NASA Goddard Space Flight Center, Jim Szykman of EPA, and the ESA-Pandonia team from Luftblick in supporting the deployment and maintenance of the Pandora instruments as well as the acquisition and processing of those observations during KORUS-AQ. We would also like to thank Ron Cohen of UC-Berkeley and his research group for their observations of $NO_2$ from the DC-8 aircraft during this same time period, and Andy Weinheimer of NCAR and his research group for their observations of $NO_y$ from the DC-8 aircraft. We would also like to thank Louisa Emmons and Gaby Pfister for their support in running the WRF-Chem simulation. Additionally, we would like to thank Jim Crawford of NASA Langley and Barry Lefer of NASA Headquarters for their input on this research article. All data from KORUS-AQ can be downloaded freely from http://www-air.larc.nasa.gov/cgi-bin/ArcView/discover-aq.dc-2011. We acknowledge the free use of $NO_2$ column data from the OMI sensor available at: https://disc.gsfc.nasa.gov/Aura/data-holdings/OMI/omno2_v003.shtml. The submitted manuscript has been created by UChicago Argonne, LLC, Operator of Argonne National Laboratory ("Argonne"). Argonne, a U.S. Department of Energy Office of Science laboratory, is operated under Contract No. DE-AC02-06CH11357. The U.S. Government retains for itself, and others acting on its behalf, a paid-up nonexclusive, irrevocable worldwide license in said article to reproduce, prepare derivative works, distribute copies to the public, and perform publicly and display publicly, by or on behalf of the Government.

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

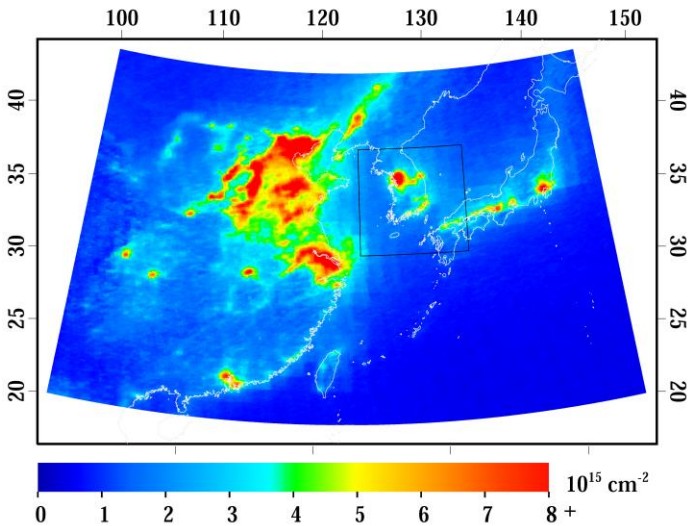

**Figure 1.** Warm season averaged (May – Sept) NO$_2$ tropospheric vertical column content using the OMI-Standard NO$_2$ product for the years of 2015 – 2017 in East Asia. The $4 \times 4$ km$^2$ WRF-Chem domain is outlined over the Korean peninsula.

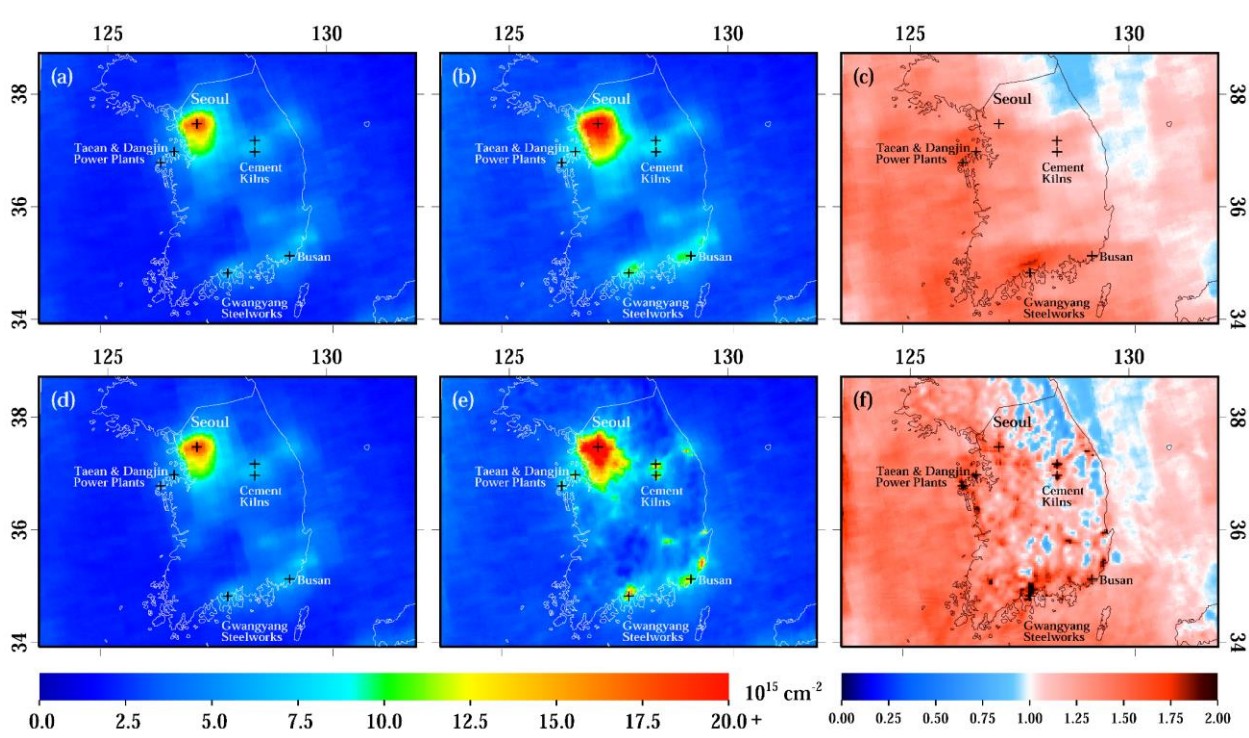

**Figure 2.** (a) OMI-Standard NO$_2$ product averaged over a 9-month period, Apr – Jun 2015 – 2017, (b) the OMI-Regional NO$_2$ product with only the air mass factor adjustment averaged over the same timeframe, and (c) the ratio between the two products. (d) Same as the top left plot, (e) the OMI-Regional NO$_2$ product with the air mass factor adjustment and spatial kernel averaged over the same timeframe, and (f) the ratio between the two products.

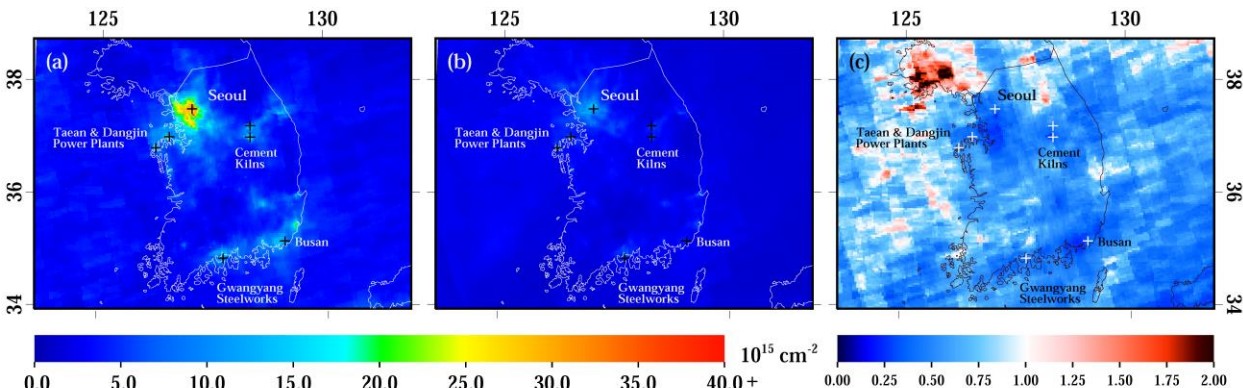

**Figure 3.** (a) The OMI-Regional NO$_2$ product with the air mass factor adjustment and spatial kernel averaged during the month of May 2016, (b) the WRF-Chem model simulation showing only days with valid OMI measurements, and (c) the ratio between the two products. On average, there are only 9 valid OMI pixels per month observed at any given location on the Korean peninsula during May 2016.

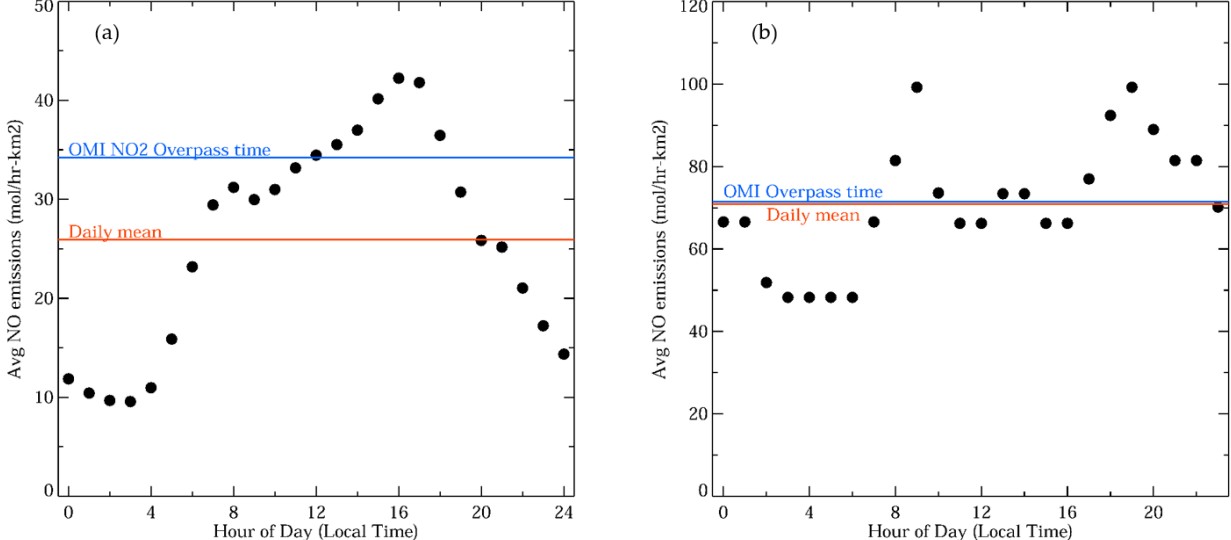

**Figure 4.** The diurnal profile of NO$_x$ emission rates processed from the bottom-up inventory. (a) The diurnal profile of NO$_x$ emission rates during a weekday in the eastern USA during July 2011 using SMOKE as the emissions pre-processor (Goldberg et al., 2016). (b) The diurnal profile of emission rates during a weekday in Korea during May 2016 using EPRES as the emissions pre-processor. Emission profiles in the right panel were used in the WRF-Chem simulation.

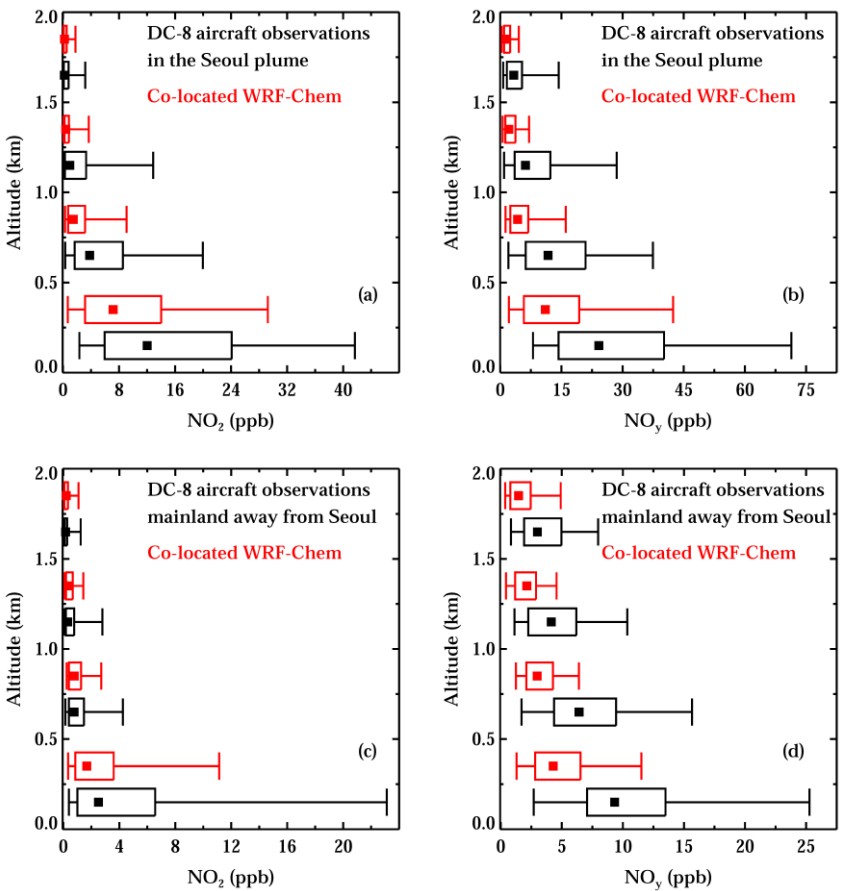

**Figure 5.** Measurements from the DC-8 aircraft binned by altitude in black. Co-located WRF-Chem within the same altitude bin as the aircraft observations are plotted above in red. Square dots represent the median values. Boxes represent the 25th and 75th percentiles, while whiskers represent the 5th and 95th percentiles. (a) Comparison of $NO_2$ in the Seoul plume (SW corner: 37.1° N, 127.05° E, NE corner: 37.75° N, 127.85° E) (b) comparison of $NO_y$ in the Seoul plume, (c) comparison of $NO_2$ in areas outside of the Seoul metropolitan area on the Korean peninsula (SW corner: 34.0° N, 126.4° E, NE corner: 37.1° N, 130.0° E), and (d) comparison of $NO_y$ in areas outside of the Seoul metropolitan area on the Korean peninsula.

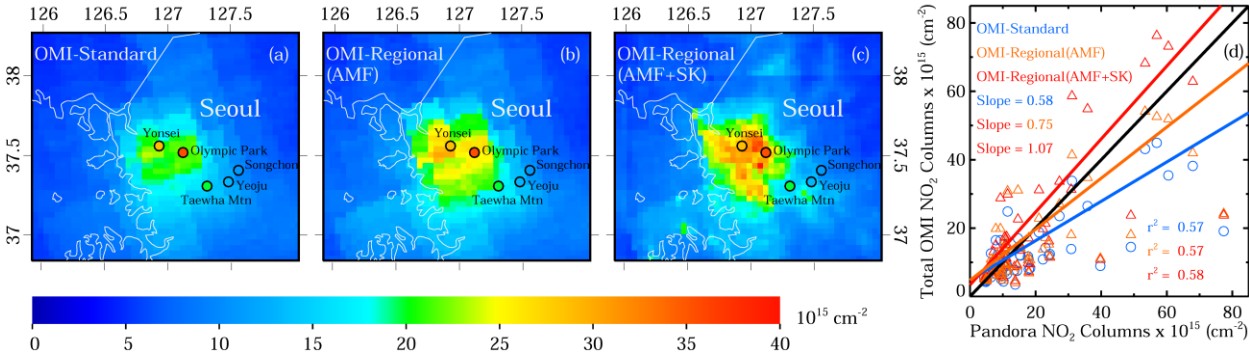

**Figure 6.** (a) Total vertical column contents from the OMI-Standard $NO_2$ product for May 2016, (b) same quantities from the OMI-Regional product with only the air mass factor adjustment (AMF) during the same timeframe, (c) same quantities from the OMI-Regional product with the air mass factor adjustment and spatial kernel (AMF+SK) during the same timeframe, and (d) a comparison between total column contents from the three OMI $NO_2$ products and Pandora $NO_2$ during May 2016. An average of Pandora 2-hour means co-located to valid daily OMI overpasses are overlaid in the spatial plots.

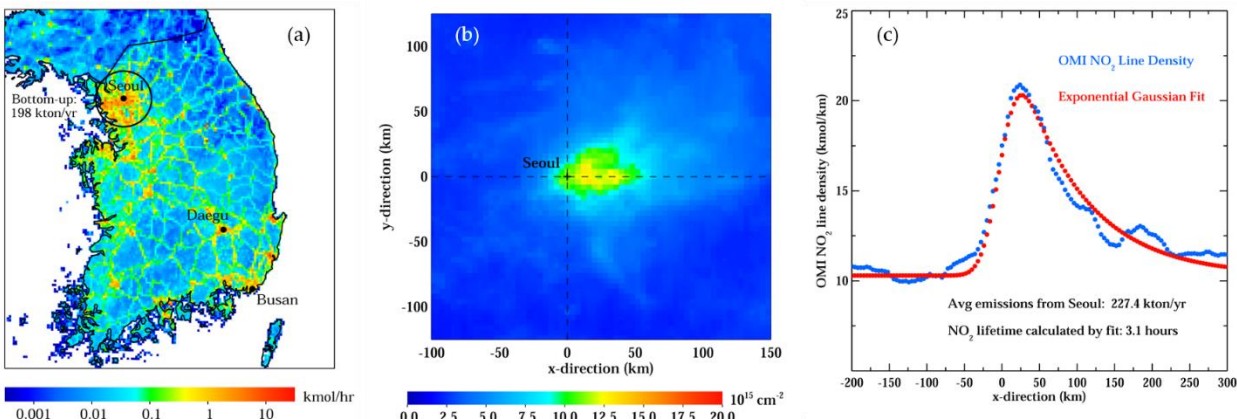

**Figure 7.** (a) Bottom-up $NO_x$ emissions inventory compiled for the KORUS-AQ field campaign, (b) the oversampled $NO_2$ plume rotated based on wind direction for Seoul, Korea from WRF-Chem ($4 \times 4$ km$^2$) for May 2016, and (c) $NO_2$ line densities integrating over the 240 km across plume width (-120 km to 120 km along the y-axis) and the corresponding EMG fit. $NO_x$ emission estimates are shown in units of kton/yr $NO_2$ equivalent and represent the mid-afternoon emissions rate.

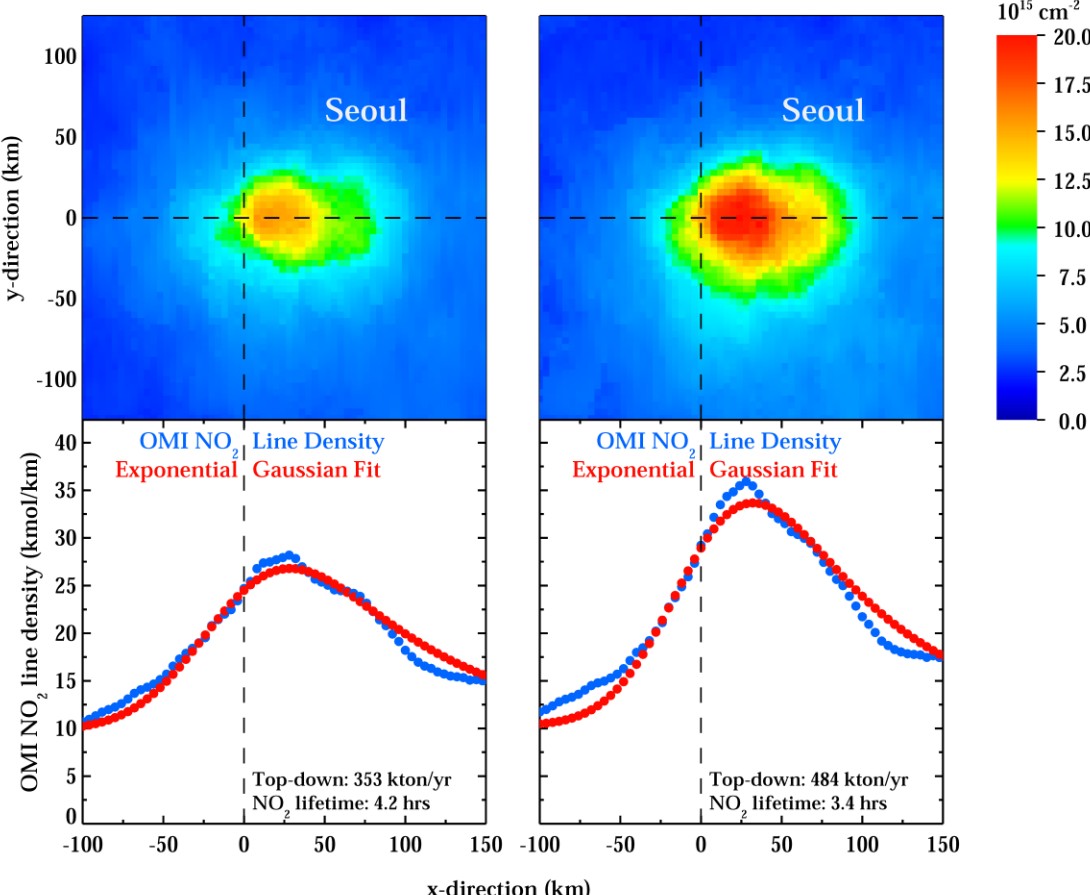

**Figure 8.** Top panels represent the oversampled ($4 \times 4$ km$^2$) OMI NO$_2$ plume from Seoul rotated based on wind direction over a 9-month period, Apr – Jun 2015 – 2017, centered on May 2016. Bottom panels represent the OMI NO$_2$ line densities integrating over the 240 km across plume width (-120 km to 120 km along the y-axis of the top panels) and the corresponding EMG fit. Left panels are from the OMI-Standard NO$_2$ product and right panels are from the OMI-Regional NO$_2$ product. NO$_x$ emission estimates are shown in units of kton/yr NO$_2$ equivalent and represent the mid-afternoon emissions rate.

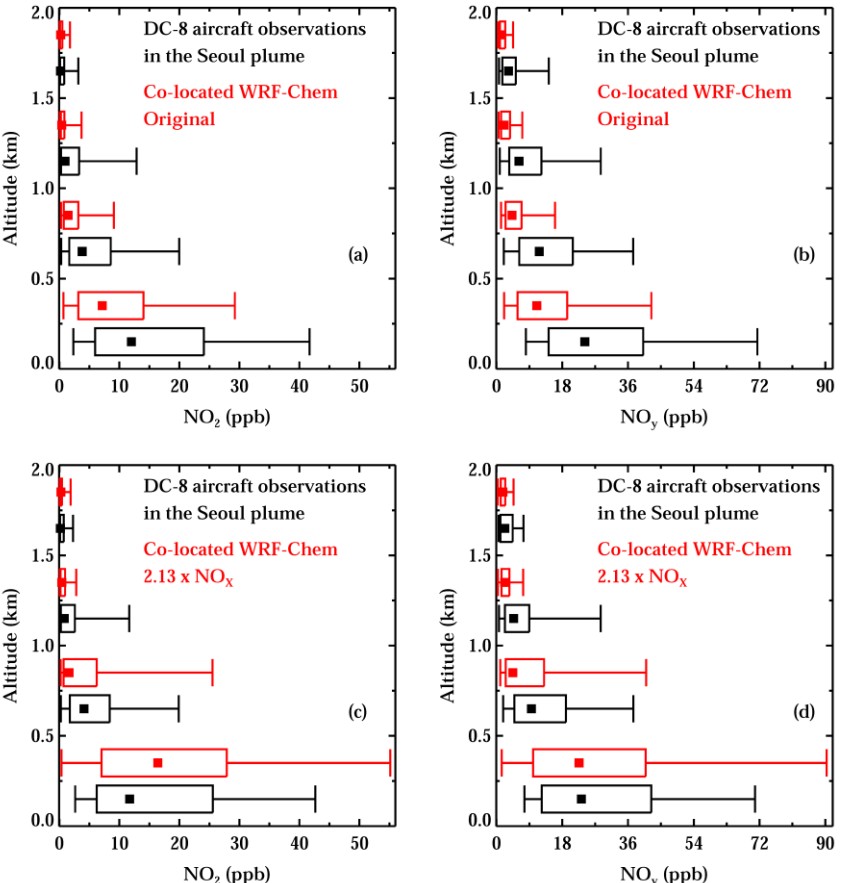

**Figure 9.** Measurements from the DC-8 aircraft binned by altitude in black. Co-located WRF-Chem within the same altitude bin as the aircraft observations are plotted above in red. Square dots represent the median values. Boxes represent the 25th and 75th percentiles, while whiskers represent the 5th and 95th percentiles. (a) Comparison of $NO_2$ in the Seoul plume (SW corner: 37.1° N, 127.05° E, NE corner: 37.75° N, 127.85° E) (b) comparison of $NO_y$ in the Seoul plume, (c) same as (a), but now using the WRF-Chem simulation with $NO_x$ emissions increased by a factor of 2.13 (d) same as (b), but now using the WRF-Chem simulation with $NO_x$ emissions increased by a factor of 2.13.

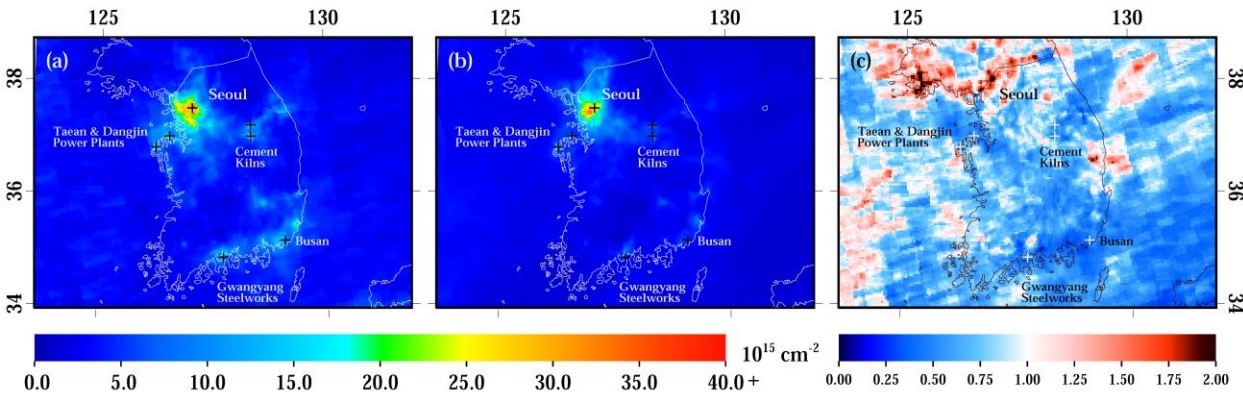

**Figure 10.** Same as Figure 3, but now showing the WRF-Chem simulation with $NO_x$ emissions in the Seoul metropolitan area increased by a factor of 2.13 in panel (b).

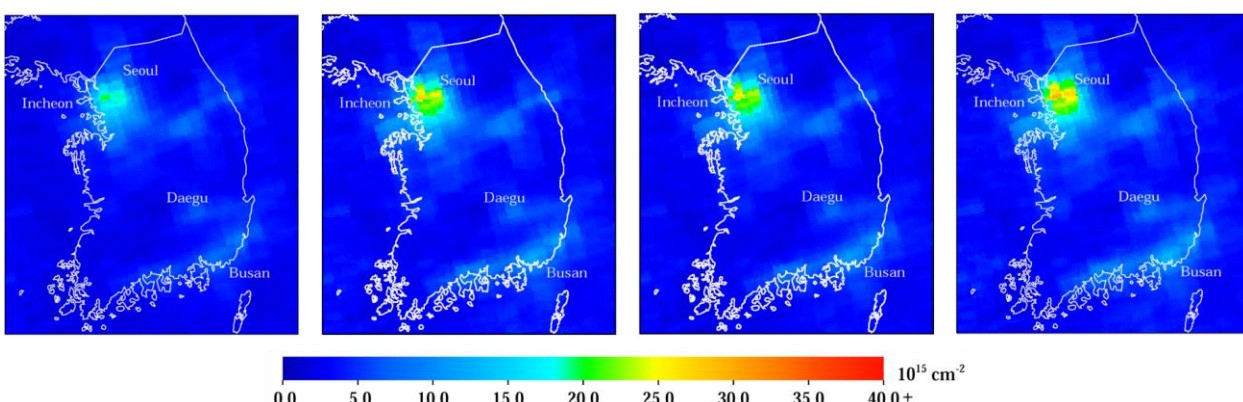

**Figure 11.** (a) The OMI-Standard product during the month of May 2016, (b) the OMI-Regional $NO_2$ product with the WRF-Chem air mass factor adjustment and spatial kernel during the same period, (c) same as (b) but using WRF-Chem $NO_2$ profiles scaled based on the aircraft comparison, and (d) same as (b) but using the WRF-Chem simulation with $NO_x$ in the Seoul metropolitan area emissions increased by a factor of 2.13.