# Peer review of "A top-down assessment using OMI NO2 suggests an underestimate in the NOx emissions inventory in Seoul, South Korea during KORUS-AQ"

_Atmospheric Chemistry and Physics, 2018_

## Referee Comment (RC1) · Anonymous Referee #1 · 17 Aug 2018

The manuscript by Goldberg et al. is a valuable and timely analysis of NOx emissions during KORUS-AQ. It identifies some potential issues with NOx emissions in the region that are useful for air quality management as well as other works studying pollution during this campaign period. The work also has relevance beyond KORUS-AQ in terms of how OMI data is used to estimate NOx from urban areas, and also how TROPOMI data will be used in such studies in the future. The article is in general quite clear and easy to read, and most figures are useful and essential.

That being said, the work misses a critical opportunity to evaluate one of their main hypotheses, which is that regionally-derived NO2 columns (using air mass factors from

high-resolution WRF-Chem simulations) lead to objectively better NOx inversions. In fact, while they report the difference between these NOx inversions and those based on the standard OMI NO2 data, the differences aren't critically evaluated, which is a shame, as it seems to be a rather easy next step. This would thus be my primary suggestion for revision. A few other aspects such as how using AMFs derived from a model that is clearly inaccurate to begin with affect their analysis, why spatial averaging is presented and then discarded, and why the regionally-derived NO2 columns may be overestimating NO2 in rural areas need to also be addressed.

Details of these comments as well as other are presented below; addressing them likely constitutes major revisions as additional WRF-Chem calculations are required.

Major comments:

Section 3.6: It isn't clear to me why the authors test a doubling of the emissions. The prior bottom-up values are 198, the top-down using standard product are 353 (an increase of x1.78) and the top-down using the regional product are 484 (an increase of x2.44). The test increase of x2 thus does little to distinguish between these two. This is a bit of a disappointment, as a major conclusion from this work is that the regional product (and top-down emissions using this product) are significantly different and better than the standard product. However, the only evidence presented that the regional product is better than standard thus far is the comparison to Pandora data. While encouraging, the authors are missing an big opportunity to make this argument much stronger by performing two model simulations for the entire KORUS-AQ period with top-down emissions that match those derived using the standard product and the regional product, precisely, and not some estimate of x2 that is neither here nor there. These two different model simulations can then be evaluated using the aircraft data.

General: Model values of NO2 column are much lower than regionally-derived OMI NO2 column in most areas, including rural areas (Fig 3). However model values match the aircraft data in rural areas (i.e. the only major discrepancies noted in discussion

of Fig 5 or e.g. the conclusions (12.17-19)). What are we thus to make then of the quality of the regionally-derived OMI values in rural areas? Too high? This should be discussed. If these are too high, will the background values estimated in the EMG value thus be too high, and this error propagate into an error in the urban emissions?

General: If model columns are too low, how does that impact model calculated AMF? How much would AMF change if using posterior emissions in WRF-Chem? An additional calculation of AMFs based on WRF-Chem simulations with adjusted emissions needs to be performed to answer this question. Or perhaps the NO2 profiles in WRF-Chem are adjusted to account for this bias (this is indicated on 4.23, but no details are provided as to what this adjustment is, or how it is derived)? I try to evaluate the WRF-Chem profiles visually, based on Fig 5, but this plot doesn't make that information clearly visible given the way the vertical axis isn't strictly used (i.e. model and aircraft data collected at the same height are not plotted at the same height – which I understand from the perspective of clarity in showing their differences with box-whisker plots, but something else is needed to evaluate profile shapes).

General: if results with spatial ave kernel are not trusted for analysis, they should be removed throughout from the results. Otherwise, it is a bit of a distracting / potentially misleading presentation. For example on page 12, line 5 – this isn't used, so why is it highlighted here? Still, wouldn't there be some data from KORUS-AQ with which wind field estimates in WRF could be evaluated? It just seems a bit subjective here that this source of error is singled out (11.18) as justification for not using this approach, whereas profile shapes that come from WRF-Chem are deemed acceptable, even though WRF-Chem NO2 column values are significantly biased low in urban areas. Further, it seems that comparison to the Pandora data in Fig 6 would indicate that the spatial kernel adjustment is improving, rather than degrading, the column estimates, which is a point in favor of this approach.

9.30-34: Not sure how this statement about NOx diurnal variability contributes to the difference between modeled and observed NO2 columns. Are the authors suggesting

that the diurnal variability of NOx emissions in Korea is incorrect? Simply noting that it is different than modeled diurnal variability in the US is not sufficient evidence and in fact comes across as tangential, unless the authors are claiming that NOx source profiles (EGUs, distribution of diesel vehicles in the transportation fleet) are identical, which seems dubious. So I suggest removing Fig 4, unless this argument can be substantially strengthened. Additionally, I wonder to what extent excessive NO2 deposition in the model might be contributing to the noted differences; this could be driven by e.g. PBL heights in the model that are too low. I suspect there is more information from the KORUS-AQ campaign that could be used to evaluate this.

Fig 5 and associated text: I agree this suggests the differences between WRF-Chem and OMI near Seoul are likely driven by emissions, rather than chemistry, deposition, or PBL heights, as suggested by the authors or myself.

10.20: Thoughts on why bias improves but not correlation? This might suggest that the daily variability of WRF-Chem (which impacts daily AMFs) is not correct, or at least not an improvement upon larger-scale averages.

General: How does the plume analysis / rotation / EMG inversion process work if e.g. there is a large point source whose outflowing plume flows over another source (e.g. a highway) that runs parallel underneath it, replenishing NO2 concentrations that are then going to be ascribed only to emissions at a single point of plume origin? So, related, at 11.10: Yes, but the concern is rather smaller sources within this radius but not at the center that contribute to the plume (i.e. mobile sources).

Minor comments and corrections:

Throughout: "shape profiles" reads a bit strange. Change to "profile shapes"? Or just profiles?

1.25: for the –> for 1.26: "larger near large" rewrite

2.4-5: "another . . . another" rewrite

3.27: trace-gas

Eq. 2: include a proper summation index

4.5-6: It isn't clear here if the authors are discussing how AMFs are calculated in general, in the standard retrieval, or in their own regionally-specific retrieval. Please clarify.

5.1: How big of an assumption is this, that the profiles are constant over this time range?

6.26: I'm pretty sure AOD from geostationary satellites over Korea have been used for forecasting studies. Not sure though how the authors here qualify their study as "near-real time"; all I saw was reanalysis. NRT usually means forecasting. Just because the winds were forecast within the domain doesn't mean this is a chemical forecast, since the observations used span the time period over which the analysis (aircraft obs) are made (considerably, given that satellite data for several more years and months are used). This entire approach would be impossible in an NRT setting, given the data requirements for oversampling.

7.28: plume, –> plume

8.6: Why using wind estimates from a different model than the one used to constrain WRF met a the boundary (NCEP), or different from WRF itself?

8.8: Why 500m? Based on Fig 5 it looks like NO2 plumes extend much higher than that, up to 1 km or possibly above (although a bit hard to tell from this plot, given the manner in which the vertical scale is treated).

Fig 1: content –> concentration

Fig 1: Why showing US domain?

9.4: is in despite of —> is despite

Section 3.1: Inclusion of / comparison to the US feels tangential and unnecessary. Suggest focus on Korea domain; remove US from Fig 1 and remove discussion here. This point could be touched on in intro or conclusions but doesn't fit well in the results.

9.17: There are also small decreases in the southern part of the peninsula, as well the SE corner of the domain. Further, the explanation provided for the decreases isn't particularly insightful.

9.21: From the presence of red in panel (c), the statement "in all areas" does not seem to accurately describe the results. Please update text to more precisely reflect the findings.

Section 3.3.1: it's not good style to have only one subsub section in a section. Consider merging this with 3.3 or making 3.3 WRF-Chem evaluation, 3.3.1 comparison to OMI and 3.3.2 comparison to aircraft.

---

## Referee Comment (RC2) · Anonymous Referee #2 · 29 Aug 2018

In "A top-down assessment using OMI NO2 suggests an underestimate in the NOx emissions inventory in Seoul, South Korea during KORUS-AQ," the authors combine two lines of research by 1) adjusting space-based retrievals of tropospheric NO2 columns with spatially refined model data and then 2) estimating NOx emissions, NOx lifetime and background tropospheric NOx by applying an Exponentially Modified Gaussian fit to the resulting NO2 column field. The authors test their method using model results that had been generate for forecasting purposes (which they determine to be successful by comparing top-down estimate to a bottom-up integration of emissions within what seems to be an arbitrary 40 km radius of Seoul).

In general, the paper is well-written and is relevant to Atmospheric Chemistry and Physics. I have two major concerns. The authors ignore impacts of topography and local circulations on the spatial gradients of the NO2 column, the quantity that determines NO2 lifetime and emissions in the analysis. Seoul is in a mountain basin terrain at the coast with further impacts on local atmospheric circulations from urban land use (e.g., https://www.atmos-chem-phys.net/13/2177/2013/acp-13-2177-2013.pdf). Also, the use of KORUS data in this manuscript for understanding the problem is limited, or is non-existent as it relates to understanding the NO2 lifetime and NOy partitioning.

The authors can use this opportunity to analyze the KORUS-AQ dataset, to compare observed NOx/NOy partitioning versus the NO2 lifetime inferred in their analysis. The authors state that the NO2 lifetime is not necessarily related to the true chemical lifetime (P12, L22-23), but the theoretical framework for the EMG method assumes that is the case, as NOx lifetime, emissions and background concentration are the only variables affecting total integrated NO2 mass. I also recommend that the authors evaluate more than two days of model-observation comparisons in the assessment of the updated CTM simulation and add to the discussion (Fig. 9, P11 L29 – P12 L2) .

This paper and the KORUS dataset provides an excellent opportunity to address the above complications and concern and I recommend the authors add more detailed analysis and discussion before publication.

Additional comments:

P1: L21-22: "regional NASA OMI NO2 product" clarify wording as this is not an official NASA product but rather is regional inputs to a NASA tropospheric SCD product

L25-26: Do the reported scalar quantities refer to integrated mass or a scalar difference at a single point?

L30-32: Consider clarifying

P2: L3: "Ideal" – this is strange wording as some ozone production occurs nearly

everywhere in the troposphere with sufficient light at wavelengths less than 405 nm. Net production is another question.

L6-7: Consider word choice. Lightning is also a source. Furthermore, there is a large difference between budget and burden, which seems to be confused here. For example, the largest contributor to the atmospheric burden of NO2 is stratospheric N2O photolysis.

L8: I recommend changing the wording "NO2 is one of the easiest trace gases to observe" to commenting that there is a rich legacy of NO2 measurements by remote sensing that has been validated.

P3: L4: Consider inclusion of Zhou et al.

Zhou, Y., D. Brunner, R. J. D. Spurr, K. F. Boersma, M. Sneep, C. Popp and B. Buchmann, Accounting for surface reflectance anisotropy in satellite retrievals of tropospheric NO2. Atmos. Meas. Tech., 3, 1185-1203, 2010.

Zhou, Y., D. Brunner, K. F. Boersma, R. Dirksen, and P. Wang, An improved tropospheric NO2 retrieval for OMI observations in the vicinity of mountaineous terrain. Atmos. Meas. Tech., 2, 401-416, 2009.

P4: L23-24: Please explain in more detail how "vertical profiles are scaled based on a comparison with in situ aircraft observations"

P5: L1: "We used May 2016 monthly mean values" Please briefly mention here whether data outside of May 2016 will be used in the top-down emissions

P6: L26-27: I am quite skeptical that this is the first time that geostationary products have been used in a forecasting framework. E.g., https://journals.ametsoc.org/doi/abs/10.1175/2008WAF2222165.1

L31: Does "enhancements" mean additions to the inventory or does it mean modifications to the inventory? Please clarify

P7 L2: Why project to 2015 and not project emissions to 2016?

P8 L8-10: This statement would be stronger if a reference is cited

L10-12: The ratio of $NO_2$ to $NO_x$ is time-dependent and spatially varying, depending primarily on $JNO_2$ and $O_3$. This should at least be noted.

L21: please clarify that you are referring to ERA-Interim winds

P9: L19-34: The chemical lifetime of $NO_x$ is impacted by uncertainties in the simulations VOC concentrations and type, OH, $RO_2$ and $R(O)O_2$ radicals. As an example, simulated atmospheric concentrations of aromatic compounds in Seoul are much smaller than observed during KORUS-AQ. An underestimate of aromatics would certainly have a large impact on peroxyacyl and alkyl nitrate formation, and should enhance the effective $NO_x$ lifetime in the near-field.

Without further justification or analysis, I don't think the comparison of emission timing is all that helpful. We would expect a different time-of-day profile of $NO_x$ emissions in Korea and USA given that the source mix of $NO_x$ is different. Emissions timing can be listed as an uncertainty without devoting an entire figure to it (Fig. 4).

P10: L20-21: please report the correlation coefficients here and refer to Figure 6.

L32: If archived, please compute the chemical lifetime of $NO_x$ in the model based on all $NO_y$ species.

P11: L1-4: From where do the uncertainty estimates affecting the analysis originate?

L3-4: "Only the latter three terms are used to calculate the uncertainty of the $NO_2$ lifetime" Why? The $NO_x$ to $NO_2$ ratio has a large impact on the $NO_2$ ($NO_x$) chemical lifetime as NO removal tends to be much slower than $NO_2$ removal.

L6-7: Why choose a radius of 40 km when the rotated plume domain width is 250 km? It seems a bit arbitrary as distance can be adjusted to improve the comparison of top-down results with bottom-up inputs.

P11: L30-31: I recommend completing a validation analysis of more than two days' worth of simulations.

P12: L24-25: "This is because the lifetime calculation is extremely sensitive to the accuracy of the wind direction." Given that top-down inference of lifetime, emissions and estimates of background NO2 are directly linked in the EMG analysis, the above statement is also true for inference of top-down emissions and background NO2. Is this "extreme sensitivity" appropriately characterized by the 30% error estimate reported on P11, L3? Please move this discussion from conclusions section on P12 to error estimate discussion early on P11 and provide a more detailed accounting.

---

## Author Response (AR1)

**Review #1**

The manuscript by Goldberg et al. is a valuable and timely analysis of NOx emissions during KORUS-AQ. It identifies some potential issues with NOx emissions in the region that are useful for air quality management as well as other works studying pollution during this campaign period. The work also has relevance beyond KORUS-AQ in terms of how OMI data is used to estimate NOx from urban areas, and also how TROPOMI data will be used in such studies in the future. The article is in general quite clear and easy to read, and most figures are useful and essential.

That being said, the work misses a critical opportunity to evaluate one of their main hypotheses, which is that regionally-derived NO2 columns (using air mass factors from high-resolution WRF-Chem simulations) lead to objectively better NOx inversions. In fact, while they report the difference between these NOx inversions and those based on the standard OMI NO2 data, the differences aren't critically evaluated, which is a shame, as it seems to be a rather easy next step. This would thus be my primary suggestion for revision. A few other aspects such as how using AMFs derived from a model that is clearly inaccurate to begin with affect their analysis, why spatial averaging is presented and then discarded, and why the regionally-derived NO2 columns may be overestimating NO2 in rural areas need to also be addressed.

Details of these comments as well as other are presented below; addressing them likely constitutes major revisions as additional WRF-Chem calculations are required.

**Thank you for your comments; they have substantially improved our manuscript.**

**Major comments:**

**V**Section 3.6: It isn't clear to me why the authors test a doubling of the emissions. The prior bottom-up values are 198, the top-down using standard product are 353 (an increase of x1.78) and the top-down using the regional product are 484 (an increase of x2.44). The test increase of x2 thus does little to distinguish between these two. This is a bit of a disappointment, as a major conclusion from this work is that the regional product (and top-down emissions using this product) are significantly different and better than the standard product. However, the only evidence presented that the regional product is better than standard thus far is the comparison to Pandora data. While encouraging, the authors are missing an big opportunity to make this argument much stronger by performing two model simulations for the entire KORUS-AQ period with top-down emissions that match those derived using the standard product and the regional product, precisely, and not some estimate of x2 that is neither here nor there. These two different model simulations can then be evaluated using the aircraft data.

In this revised manuscript, we have completed a month-long simulation with NOx emissions increased by a factor of 2.13, and have removed the two-day 2 × NOx scenario. A factor of 2.13 is chosen because the top-down estimate from the satellite is 484 kton/yr, while the top-down approach applied to the model is 227 kton/yr. The bottom-up NOx emissions inventory within a 40 km radius of Seoul is 198 kton/yr, however the 227 kton/yr value is a more appropriate comparison with the top-down satellite analysis.

We are confident the OMI-Regional NO2 product is more robust than the standard product due to the comparison with the Pandora NO2 network. Furthermore, the methodology of updating the satellite

product with high-resolution a priori NO2 shape profiles is more scientifically appropriate for regional studies (Russell et al., 2012, Lamsal et al., 2015, Kuhlmann et al., 2016, Goldberg et. al., 2017).

Thus, we feel that it is unnecessary to perform a simulation with NOx increased by a factor of 1.56 (353 kton/yr vs. 227 kton/yr). Furthermore, as we show in two new figures, the updated 2.13 x NOx simulation agrees well with the aircraft data (Figure 9) and the OMI-Regional NO2 product (Figure 10).

**V**General: Model values of NO2 column are much lower than regionally-derived OMI NO2 column in most areas, including rural areas (Fig 3). However model values match the aircraft data in rural areas (i.e. the only major discrepancies noted in discussion of Fig 5 or e.g. the conclusions (12.17-19)). What are we thus to make then of the quality of the regionally-derived OMI values in rural areas? Too high? This should be discussed. If these are too high, will the background values estimated in the EMG value thus be too high, and this error propagate into an error in the urban emissions?

In the original figure, we are referring to the "mainland transect". This is a subset of the rural areas, and was inappropriate. We have since updated the figure to include all mainland areas away from Seoul, and now find a discrepancy between NO2 in the lowest layers between the model and the aircraft observations. This figure and corresponding discussion has been updated.

**V**General: If model columns are too low, how does that impact model calculated AMF? How much would AMF change if using posterior emissions in WRF-Chem? An additional calculation of AMFs based on WRF-Chem simulations with adjusted emissions needs to be performed to answer this question.

A new figure, Figure 11, now addresses this. The effect of the emissions inventory on the air mass factor is appreciable, but is secondary to the resolution of the model simulation. In the Seoul metropolitan area, the AMF changes on average by 35% when switching from GMI to WRF-Chem and changes by only 8% when switching emission inventories.

**V**Or perhaps the NO2 profiles in WRFChem are adjusted to account for this bias (this is indicated on 4.23, but no details are provided as to what this adjustment is, or how it is derived)? I try to evaluate the WRF-Chem profiles visually, based on Fig 5, but this plot doesn't make that information clearly visible given the way the vertical axis isn't strictly used (i.e. model and aircraft data collected at the same height are not plotted at the same height – which I understand from the perspective of clarity in showing their differences with box-whisker plots, but something else is needed to evaluate profile shapes).

The OMI-Regional NO2 product derived herein already accounts for any mean model biases. A better description of this process is now provided in Section 2.1.1.

**V**General: if results with spatial ave kernel are not trusted for analysis, they should be removed throughout from the results. Otherwise, it is a bit of a distracting / potentially misleading presentation. For example on page 12, line 5 – this isn't used, so why is it highlighted here? Still, wouldn't there be some data from KORUS-AQ with which wind field estimates in WRF could be evaluated? It just seems a bit subjective here that this source of error is singled out (11.18) as justification for not using this approach, whereas profile shapes that come from WRF-Chem are deemed acceptable, even though WRF-Chem NO2 column values are significantly biased low in urban areas. Further, it seems that

comparison to the Pandora data in Fig 6 would indicate that the spatial kernel adjustment is improving, rather than degrading, the column estimates, which is a point in favor of this approach.

As noted, the spatial averaging kernel provides important insight into resolving discrepancies between OMI NO2 and Pandora NO2. However, we also emphasize that the spatial averaging kernel has its limitations. The top-down approach is extremely sensitive to wind direction, so any errors in the forecasted wind fields will propagate through to the top-down method. When we apply a spatial averaging kernel to the satellite retrieval and then perform the top-down method, a NOx emissions rate cannot be derived. Therefore, for the top-down analysis, the artificial error introduced by spatial averaging kernel outweighs its benefits. However, for the Pandora comparison, the benefits outweigh the artificial errors (as shown in Figure 6).

**V**9.30-34: Not sure how this statement about NOx diurnal variability contributes to the difference between modeled and observed NO2 columns. Are the authors suggesting that the diurnal variability of NOx emissions in Korea is incorrect? Simply noting that it is different than modeled diurnal variability in the US is not sufficient evidence and in fact comes across as tangential, unless the authors are claiming that NOx source profiles (EGUs, distribution of diesel vehicles in the transportation fleet) are identical, which seems dubious. So I suggest removing Fig 4, unless this argument can be substantially strengthened.

We are suggesting that the temporalization of NOx emissions can introduce errors in satellite and aircraft measurements, which occur during the daytime. The temporalization is a best estimate based on literature, but it is almost certainly not correct either. The temporalization of NOx emissions as a major source of the discrepancy has not been discussed in previous literature and is quite critical to the conclusions of this manuscript. Resolving these differences is an important topic for future research.

However, we are not necessarily suggesting that the Korean temporalization is identical to the eastern US, but instead are providing a comparison to show how temporalization can differ by region.

The discussion of this topic in the text has been added to and is now referenced in the Conclusions as an important source of the discrepancy.

**V**Additionally, I wonder to what extent excessive NO2 deposition in the model might be contributing to the noted differences; this could be driven by e.g. PBL heights in the model that are too low. I suspect there is more information from the KORUS-AQ campaign that could be used to evaluate this.

We have now included a comparison with  $NO_y$ . Evaluating the  $NO_2$  deposition rates and PBL heights is beyond the scope of this study.

**V**Fig 5 and associated text: I agree this suggests the differences between WRF-Chem and OMI near Seoul are likely driven by emissions, rather than chemistry, deposition, or PBL heights, as suggested by the authors or myself.

**V**10.20: Thoughts on why bias improves but not correlation? This might suggest that the daily variability of WRF-Chem (which impacts daily AMFs) is not correct, or at least not an improvement upon larger-scale averages.

Yes, these are our thoughts too. The WRF simulation used to drive the chemistry is in forecast mode. This has been clarified in the text.

**V**General: How does the plume analysis / rotation / EMG inversion process work if e.g. there is a large point source whose outflowing plume flows over another source (e.g. a highway) that runs parallel underneath it, replenishing NO2 concentrations that are then going to be ascribed only to emissions at a single point of plume origin? So, related, at 11.10: Yes, but the concern is rather smaller sources within this radius but not at the center that contribute to the plume (i.e. mobile sources).

Small sources at the edge of the urban boundary will lead to an artificially longer NO2 lifetime. This partially compensates the error introduced by the wind. A short commentary has now been included in the Section 3.6.2 of the manuscript.

**Minor comments and corrections:**

**V**Throughout: "shape profiles" reads a bit strange. Change to "profile shapes"? Or just profiles?

**Updated**

**V**1.25: for the -> for 1.26: "larger near large" rewrite

**Updated**

**√**2.4-5: "another . . . another" rewrite

**Updated**

**V**3.27: trace-gas Eq. 2: include a proper summation index

**Updated**

**V**4.5-6: It isn't clear here if the authors are discussing how AMFs are calculated in general, in the standard retrieval, or in their own regionally-specific retrieval. Please clarify.

It is in reference to all OMI NO2 products derived from the NASA OMI NO2 product. This includes both the standard product and the regional product derived here (as well as any other custom products derived from the NASA product). It has been clarified.

V5.1: How big of an assumption is this, that the profiles are constant over this time range?

Please reference Laughner et al., 2016, which is already cited here. That study shows that the AMF can vary by 20% on a daily basis.

**V**6.26: I'm pretty sure AOD from geostationary satellites over Korea have been used for forecasting studies.

The sentence referring to this simulation as the first near real-time application of geostationary data has been removed.

**V**6.26: Not sure though how the authors here qualify their study as "nearreal time"; all I saw was reanalysis. NRT usually means forecasting. Just because the winds were forecast within the domain doesn't mean this is a chemical forecast, since the observations used span the time period over which the analysis (aircraft obs) are made (considerably, given that satellite data for several more years and months are used). This entire approach would be impossible in an NRT setting, given the data requirements for oversampling.

This statement is in reference to the model simulation only. The model simulation was indeed performed as a forecast in near-real time. The OMI NO2 satellite data was processed after the fact, but AOD was in fact assimilatiated in near-real time.

**√**7.28: plume, -> plume**

**Updated**

**V**8.6: Why using wind estimates from a different model than the one used to constrain WRF met at the boundary (NCEP), or different from WRF itself?

The WRF simulation is a forecast simulation. Re-analysis data is more robust despite it being at a coarser spatial resolution.

**V**8.8: Why 500m? Based on Fig 5 it looks like NO2 plumes extend much higher than that, up to 1 km or possibly above (although a bit hard to tell from this plot, given the manner in which the vertical scale is treated).

We follow Lu et al., 2015. Generally, winds do not vary much between 500 – 1000 m. De Foy et al., 2014 discuss how the selection of wind speeds/direction affect the top-down calculation. This is taken into account in the uncertainty analysis.

**√**Fig 1: content –> concentration**

The word "content" is correct in this context. Concentration is mass per unit volume, which is not being shown here.

**√**Fig 1: Why showing US domain?

This has now been removed, but the US is still referenced in the text for comparison.

**√**9.4: is in despite of --> is despite

**Updated**

**V**Section 3.1: Inclusion of / comparison to the US feels tangential and unnecessary. Suggest focus on Korea domain; remove US from Fig 1 and remove discussion here. This point could be touched on in intro or conclusions but doesn't fit well in the results.

The US figure has now been removed, but the US is still quickly referenced in the text of this section for comparison.

 $\mathbf{V}$ 9.17: There are also small decreases in the southern part of the peninsula, as well the SE corner of the domain. Further, the explanation provided for the decreases isn't particularly insightful.

This sentence has been removed.

 $\mathbf{V}$ 9.21: From the presence of red in panel (c), the statement "in all areas" does not seem to accurately describe the results. Please update text to more precisely reflect the findings.

The word "all" has been changed to "most"

**V**Section 3.3.1: it's not good style to have only one subsub section in a section. Consider merging this with 3.3 or making 3.3 WRF-Chem evaluation, 3.3.1 comparison to OMI and 3.3.2 comparison to aircraft.

This section is now a section by itself, since it is now expanded.

**Review #2**

In "A top-down assessment using OMI NO2 suggests an underestimate in the NOx emissions inventory in Seoul, South Korea during KORUS-AQ," the authors combine two lines of research by 1) adjusting spacebased retrievals of tropospheric NO2 columns with spatially refined model data and then 2) estimating NOx emissions, NOx lifetime and background tropospheric NOx by applying an Exponentially Modified Gaussian fit to the resulting NO2 column field. The authors test their method using model results that had been generate for forecasting purposes (which they determine to be successful by comparing topdown estimate to a bottom-up integration of emissions within what seems to be an arbitrary 40 km radius of Seoul).

**V**In general, the paper is well-written and is relevant to Atmospheric Chemistry and Physics.

Thank you for your comments; they have substantially improved our manuscript.

**V**I have two major concerns. The authors ignore impacts of topography and local circulations on the spatial gradients of the NO2 column, the quantity that determines NO2 lifetime and emissions in the analysis. Seoul is in a mountain basin terrain at the coast with further impacts on local atmospheric circulations from urban land use (e.g., https://www.atmos-chem-phys.net/13/2177/2013/acp-13-2177-2013.pdf).

We agree that it is important to reference the complex topography and meteorology of the area as sources of uncertainty, but we do not expect this source of uncertainty to bias our results in any particular direction. We have added a paragraph in the discussion section and now reference the aforementioned study and others.

The complex geography of the region further supports the use of our 4 x 4 km2 simulation because it will capture topography and mesoscale phenomena better than a coarse global model.

When re-processing the air mass factor we use surface pressure of the WRF-Chem simulation to process the air mass factor, so we are already accounting for topographical differences in surface pressures. This is already discussed in Section 2.1.1.

**V**Also, the use of KORUS data in this manuscript for understanding the problem is limited, or is nonexistent as it relates to understanding the NO2 lifetime and NOy partitioning. The authors can use this opportunity to analyze the KORUS-AQ dataset, to compare observed NOx/NOy partitioning versus the NO2 lifetime inferred in their analysis. The authors state that the NO2 lifetime is not necessarily related to the true chemical lifetime (P12, L22-23), but the theoretical framework for the EMG method assumes that is the case, as NOx lifetime, emissions and background concentration are the only variables affecting total integrated NO2 mass.

We have now included a comparison to  $NO_y$  from the DC-8 aircraft. This is now shown in Figures 5 & 9. The large underestimate of  $NO_y$  further supports the conclusions of our manuscript and makes it stronger.

It is beyond the scope of this study to do a full analysis of the NOy partitioning.

 $\mathbf{V}$ I also recommend that the authors evaluate more than two days of model-observation comparisons in the assessment of the updated CTM simulation and add to the discussion (Fig. 9, P11 L29 – P12 L2).

As suggested, a CTM simulation with 2.13 × NOx for the entire month of May 2016 is now included.

This paper and the KORUS dataset provides an excellent opportunity to address the above complications and concern and I recommend the authors add more detailed analysis and discussion before publication.

**Additional comments:**

**√**P1: L21-22: "regional NASA OMI NO2 product" clarify wording as this is not an official NASA product but rather is regional inputs to a NASA tropospheric SCD product

**Updated**

VL25-26: Do the reported scalar quantities refer to integrated mass or a scalar difference at a single point?

The scalar quantity for Seoul is within a 40 km radius. This has been clarified in Section 3.2, which is the section this statement is referencing.

**√**L30-32: Consider clarifying**

 $\mathbf{V}$ P2: L3: "Ideal" – this is strange wording as some ozone production occurs nearly everywhere in the troposphere with sufficient light at wavelengths less than 405 nm. Net production is another question.

Re-phrased to say, "In the presence of abundant volatile organic compounds and strong sunlight, NOx can participate in a series of chemical reactions to accelerate the production of O3".

**V**L6-7: Consider word choice. Lightning is also a source. Furthermore, there is a large difference between budget and burden, which seems to be confused here. For example, the largest contributor to the atmospheric burden of NO2 is stratospheric N2O photolysis.

Re-phrased to say, "There are some biogenic emissions of NOx (e.g., lightning), but the majority of the NOx emissions are from anthropogenic sources".

**V**L8: I recommend changing the wording "NO2 is one of the easiest trace gases to observe" to commenting that there is a rich legacy of NO2 measurements by remote sensing that has been validated.

**This paragraph has been re-worded as suggested.**

**V**P3: L4: Consider inclusion of Zhou et al. Zhou, Y., D. Brunner, R. J. D. Spurr, K. F. Boersma, M. Sneep, C. Popp and B. Buchmann, Accounting for surface reflectance anisotropy in satellite retrievals of tropospheric NO2. Atmos. Meas. Tech., 3, 1185-1203, 2010. Zhou, Y., D. Brunner, K. F. Boersma, R. Dirksen, and P. Wang, An improved tropospheric NO2 retrieval for OMI observations in the vicinity of mountaineous terrain. Atmos. Meas. Tech., 2, 401-416, 2009.

**Included**

 $\mathbf{V}$ P4: L23-24: Please explain in more detail how "vertical profiles are scaled based on a comparison with in situ aircraft observations"

An extra sentence has been added: "For example, if the aircraft observations show that  $NO_2$  concentrations between 0 - 500 m are low by 50%, then we scale the modeled  $NO_2$  in this altitude bin by this same amount."

As shown now shown in Figure 11, this has a minimal effect on the calculation of vertical tropospheric column contents over the Korean peninsula.

 $\mathbf{V}$ P5: L1: "We used May 2016 monthly mean values" Please briefly mention here whether data outside of May 2016 will be used in the top-down emissions

Yes, these data are also included in the top-down analysis. A sentence has been added: "In the topdown emissions derivation, we use all nine-months of OMI data for the analysis."

**V**P6: L26-27: I am quite skeptical that this is the first time that geostationary products have been used in a forecasting framework. E.g., https://journals.ametsoc.org/doi/abs/10.1175/2008WAF2222165.1

This sentence has been removed.

 $\mathbf{V}$ L31: Does "enhancements" mean additions to the inventory or does it mean modifications to the inventory? Please clarify

The main enhancement that we made is to add new construction of power plants. Per your question, "addition" would be the right word. This is clarified in the text.

**√**P7 L2: Why project to 2015 and not project emissions to 2016?

The NIER (National Institute of Environmental Research) of Korea generates a "Present Version Inventory," for their air quality forecasting, by projecting the base year inventory for three years (i.e. 2015). In this simulation, we use that version of the inventory. We have clarified that NIER provided the projected emissions, and that we did not project the emissions.

 $\mathbf{V}$ P8 L8-10: This statement would be stronger if a reference is cited

Reference to de Foy et al. (2014) is now included.

**V**L10-12: The ratio of NO2 to NOx is time-dependent and spatially varying, depending primarily on JNO2 and O3. This should at least be noted.

**Updated**

**V**L21: please clarify that you are referring to ERA-Interim winds

**Clarified**

**V**P9: L19-34: The chemical lifetime of NOx is impacted by uncertainties in the simulations VOC concentrations and type, OH, RO2 and R(O)O2 radicals. As an example, simulated atmospheric concentrations of aromatic compounds in Seoul are much smaller than observed during KORUS-AQ. An underestimate of aromatics would certainly have a large impact on peroxyacyl and alkyl nitrate formation, and should enhance the effective NOx lifetime in the near-field.

This has been updated. Romer et al., (2016) has now been cited to support this.

**V**Without further justification or analysis, I don't think the comparison of emission timing is all that helpful. We would expect a different time-of-day profile of NOx emissions in Korea and USA given that the source mix of NOx is different. Emissions timing can be listed as an uncertainty without devoting an entire figure to it (Fig. 4).

The emission timing is critical to the conclusion of this paper. We are suggesting that the timing of emissions can yield a large amount of uncertainty when evaluating emission inventories with satellite data. Thus, we still include this figure.

**√**P10: L20-21: please report the correlation coefficients here and refer to Figure 6.

Included

VL32: If archived, please compute the chemical lifetime of NOx in the model based on all NOy species.

This was not archived.

✔P11: L1-4: From where do the uncertainty estimates affecting the analysis originate?

Please reference Lu et al., (2015). We have added the citation here.

**V**L3-4: "Only the latter three terms are used to calculate the uncertainty of the NO2 lifetime" Why? The NOx to NO2 ratio has a large impact on the NO2 (NOx) chemical lifetime as NO removal tends to be much slower than NO2 removal.

Please reference Lu et al., (2015). We have added the citation here.

**V**L6-7: Why choose a radius of 40 km when the rotated plume domain width is 250 km? It seems a bit arbitrary as distance can be adjusted to improve the comparison of top-down results with bottom-up inputs.

An assumption with this method is that all of the NOx emissions are clustered near a single point. The radius of 40 km from the city center is chosen because it encompasses an area which includes the highest NOx emission sources, but very little of the emissions from more rural areas, which are contributing to the background NO2. A radius much larger than 40 km would be inappropriate.

For the calculation of the OMI line densities, we apply a 120 km radius (we are unsure where you saw the number 250 as it is not in the original manuscript). By doing so, we are assuming that emissions between a radius of 40 km and 120 km are contributing to the background. This is an assumption of the top-down method. Figure 7 confirms that this is a valid assumption.

**√**P11: L30-31: I recommend completing a validation analysis of more than two days' worth of simulations.

This is now included. Please refer to Figures 9 & 10.

**V**P12: L24-25: "This is because the lifetime calculation is extremely sensitive to the accuracy of the wind direction." Given that top-down inference of lifetime, emissions and estimates of background NO2 are directly linked in the EMG analysis, the above statement is also true for inference of top-down emissions and background NO2. Is this "extreme sensitivity" appropriately characterized by the 30% error estimate reported on P11, L3? Please move this discussion from conclusions section on P12 to error estimate discussion early on P11 and provide a more detailed accounting.

As suggested, this has been moved to Section 3.6.2. A reference is now also cited, de Foy et al. (2014), which describes in detail the uncertainty analysis.

**A top-down assessment using OMI NO2 suggests an underestimate in the NOx emissions inventory in Seoul, South Korea during KORUS-AQ**

Daniel L. Goldberg\*,1,2, Pablo E. Saide3, Lok N. Lamsal4,5, Benjamin de Foy6, Zifeng Lu1,2, Jung-Hun Woo7, Younha Kim7, Jinseok Kim7, Meng Gao8, Gregory Carmichael9, and David G.

5 Hun Woo7, Younha Kim7, Jinseok Kim7, Meng Gao8, Gregory Carmichael9, and David G. Streets1,2

1Energy Systems Division, Argonne National Laboratory, Argonne, IL 60439 USA
 2Consortium for Advanced Science and Engineering, University of Chicago, Chicago, IL 60637, USA
 3Department of Atmospheric and Oceanic Sciences, Institute of the Environment and Sustainability, University of California – Los Angeles, Los Angeles, CA 90095, USA
 4Goddard Earth Sciences Technology and Research, Universities Space Research Association, Columbia, MD 21046, USA

5NASA Goddard Space Flight Center, Code 614, Greenbelt, MD 20771, USA

6Department of Earth and Atmospheric Sciences, Saint Louis University, St. Louis, MO 63108, USA
 7Konkuk University, 05029 Seoul, South Korea
 8School of Engineering and Applied Sciences, Harvard University, Cambridge, MA 02138, USA
 9Department of Chemical and Biochemical Engineering, University of Iowa, Iowa City, IA 52242, USA
 *Correspondence to*: Daniel L. Goldberg (dgoldberg@anl.gov)

20

Abstract. In this work, we investigate the NOx emissions inventory in Seoul, South Korea using a regional NASA Ozone Monitoring Instrument (OMI) NO2 product derived from the standard NASA product. We first develop a regional OMI NO2 product by re-calculating the air mass factors using a high-resolution (4 × 4 km2) WRF-Chem model simulation, which better captures the NO2 shape profilesprofile shapes in urban regions. We then apply a model-derived spatial averaging kernel to further downscale the retrieval and account for the sub-pixel variability. These two modifications yield OMI NO2 values in the regional product that are 1.37 larger in the Seoul metropolitan region and >2 times larger near large industrialsubstantial point sources. These two modifications also yield an OMI NO2 product that is in better agreement with the Pandora NO2 spectrometer measurements acquired during the Korea U.S.-Air Quality (KORUS-AQ) field campaign. NOx emissions are then derived for the Seoul metropolitan area during the KORUS-AQ field campaign using a top-down approach with the standard and regional NASA OMI NO2

- 30 during the KORUS-AQ field campaign using a top-down approach with the standard and regional NASA OMI NO2 products. We first apply the top-down approach to a model simulation to ensure that the method is appropriate: the WRF-Chem simulation utilizing the bottom-up emission inventory yields a NOx emission rate of  $227 \pm 94$  kton/yr, while the bottom-up inventory itself yields a NOx emission rate of 198 kton/yr. Using the top-down approach on the regional OMI NO2 product, we derive the NOx emissions rate from Seoul to be  $484 \pm 201$  kton/yr, and a  $353 \pm 146$
- 35 kton/yr NOx emissions rate using the standard NASA OMI NO2 product. This suggests an underestimate of 53% and 36% using the regional and standard NASA OMI NO2 products respectively. To supplement this finding, we compare the NO2 and NOy simulated by WRF-Chem to observations of the same quantity acquired by aircraft and find a model underestimate. When NOx emissions in the WRF-Chem model are doubledincreased by a factor of 2.13 in the Seoul metropolitan area, there is better agreement with KORUS-AQ aircraft observations and the re-calculated OMI NO2
- 40 tropospheric columns. Finally, we show that by using a WRF-Chem simulation with an updated emissions inventory to re-calculate the AMF, there are small differences (10 - 20~8%) in OMI NO2 compared to using the original WRF-Chem simulation to derive the AMF. This suggests that changes in model resolution have a larger effect on the AMF

calculation than modifications to the Korean emissions inventory. Although the current work is focused on South Korea using OMI, the methodology developed in this work can be applied to other world regions using TROPOMI and future satellite datasets (e.g., GEMS and TEMPO) to produce high-quality region-specific top-down NOx emission estimates.

**1** Introduction**

Nitrogen oxides (NOx  $\equiv$  NO+NO2) are a group of reactive trace gases that are toxic to human health and can transform in the atmosphere into other noxious chemical species. During ideal atmospheric conditionsIn the presence of abundant volatile organic compounds and3 strong sunlight, NOx2 can <del>photolyze participate in a series of chemical</del>

- 5 reactions to create accelerate the production of  $O_3$ , another toxic air pollutant with a longer atmospheric lifetime.  $NO_x$  can also transform into particulate nitrate, a component of fine particulate matter ( $PM_{2.5}$ ), another an additional health hazard. When fully oxidized in the atmosphere,  $NO_x$  transforms into HNO3, a key contributor to acid rain. There are some biogenic sources emissions of  $NO_x$  (e.g., lightning), but the majority of the  $NO_{x^2}$  in our atmosphere today is emitted by emissions are from anthropogenic sources (van Vuuren et al., 2011).
- 10 There is a rich legacy of NO2 measurements by remote sensing instruments (Burrows et al., 1999). NO2-is one of the easiest trace gases to observe from space because it has strong absorption features within the 400 465 nm wavelength region (Vandaele et al., 1998). One of these instruments is Tthe Dutch-Finnish Ozone Monitoring Instrument (OMI), which measures the absorption of solar backscatter in the UV-visible spectral range. NO2 can be observed from space because it has strong absorption features within the 400 465 nm wavelength region (Vandaele et al., 1998). By
- 15

comparing observed spectra with a reference spectrum, the amount of  $NO_2$  in the atmosphere between the instrument in low-earth orbit and the surface can be derived; this technique is called differential optical absorption spectroscopy (DOAS) (Platt, 1994).

Tropospheric NO2 column contents from OMI have been used to estimate NOx emissions from various areas around the globe (Streets et al., 2013; Miyazaki et al., 2017) including North America (Boersma et al., 2008; Lu et al., 2015),  $A = (71)^{-1}$

- Asia (Zhang et al., 2008; Han et al, 2015; Kuhlmann et al., 2015; Liu et al., 2017), the Middle East (Beirle et al., 2011), and Europe (Huijnen et al., 2010; Curier et al., 2014). It has also been used to produce and validate NOx emission estimates from sectors such as soil (Hudman et al., 2010; Vinken et al., 2014a; Rasool et al., 2016), lightning (Allen et al., 2012; Liaskos et al., 2015; Pickering et al., 2016; Nault et al., 2017), power plants (de Foy et al., 2015), aircraft (Pujadas et al., 2011), marine vessels (Vinken et al., 2014b; Boersma et al., 2015), and urban centers (Lu et al., 2015; Canty et al., 2015; Souri et al., 2016).
  - With a pixel resolution varying from  $13 \times 24$  km2 to  $26 \times 128$  km2, the OMI sensor was developed for global to regional scale studies rather than for individual urban areas. Even at the highest spatial resolution of  $13 \times 24$  km2, the sensor has difficulty observing the fine structure of NO2 plumes at or near the surface (e.g., highways, power plants, factories, etc.) (Chen et al., 2009; Ma et al., 2013; Flynn et al., 2014), which are often less than 10 km in width (Heue
- 30 et al., 2008). This can lead to a spatial averaging of pollution (Hilboll et al., 2013). A temporary remedy, until higher spatial resolution satellite instruments are operational, is to use a regional air quality simulation to estimate the sub-pixel variability of OMI pixels. Kim et al. (2016) utilize the spatial variability in a regional air quality model to spatially downscale OMI NO2 measurements using a spatial averaging kernel. The spatial averaging kernel technique has shown to increase the OMI NO2 signal within urban areas, which is in better agreement with observations in these
- 35 regions (Goldberg et al., 2017).

Furthermore, the air mass factor and surface reflectance used in obtaining the global OMI NO2 retrievals are at a coarse spatial resolution (Lorente et al. 2017; Kleipool et al., 2008). While appropriate for a global operational retrieval, this is known to cause an underestimate in the OMI NO2 signal in urban regions (Russell et al., 2011). The air mass factors in operational OMI NO2 are calculated using NO2 shape profiles profile shapes that are provided at a

[revised manuscript text omitted]

---

## Author Response (AR2)

**Please see our responses below which are highlighted in red.**

**Review #1**

In general, the authors have satisfactorily addressed reviewer comments. I recommend that the authors consider the below comments and also recommend that the authors review the manuscript in careful detail to ensure that the manuscript uses accurate and precise language on specific details (E.g., along the lines as described in the paragraph below for the Introductory paragraph)

Thank you for your comments.

The first paragraph of the introduction should be edited for clarity and accuracy. 1) NO reaction with HO2 or RO2 does not "accelerate" O3 production, it is the only major known reacting leading to net production of NO2 and subsequently O3 in the troposphere. 2) lightning NOx is not biogenic, but geogenic. 3) NOx does not 'transform' in to HNO3, but rather NO2 reacts with OH or N2O5 reacts in aqueous solution to form HNO3.

These have been clarified. The revised paragraph is below:

"Nitrogen oxides (NOX ≡ NO+NO2) are a group of reactive trace gases that are toxic to human health and can be converted in the atmosphere into other noxious chemical species.  In the presence of abundant volatile organic compounds and strong sunlight, NOX can participate in a series of chemical reactions to generate a net accumulation of O3, another toxic air pollutant with a longer atmospheric lifetime.  NOX also participates in a series of reactions to create HNO3, a key contributor to acid rain, and particulate nitrate (NO3-), a component of fine particulate matter (PM2.5), an additional health hazard.  There are some natural emissions of NOX (e.g., lightning), but the majority of the NOX emissions are from anthropogenic sources (van Vuuren et al., 2011)."

The authors indicated in their response that there was no NOy simulation results archived (comment for P10L32), for the purpose of analyzing NOx lifetime in the simulation, but they added a comparison of model output to NOy observations in the revised manuscript. Please, consider a comparison of NOx:NOy in model as well as has been shown for NOx:NOy in observations.

We misunderstood the original comment as a request for data from the process analysis tool, which was not output in this simulation. We have added the $NO_2$:NOy model vs. observation comparison, and re-formatted Figure 9. Both simulations captured the $NO_2$:NOy ratio fairly well; neither simulation performed any better than the other. A sentence on this is now included in Section 3.7 and shown below:

"The NO2-NOy partitioning is captured well by both model simulations, and there is no significant change in the NO2-NOy ratio when using increased NOX emissions."

P13 L19 – L30 of annotated revised manuscript: The discussion of NO2 lifetime should edited for clarity before publication. To be an accurate means of inverting NOx emissions in the EMG method, the effective lifetime derived from NO2 column measurements must be directly, physically related to the PBL NOx lifetime (The EMG method accounts for background NO2 and NO2:NOx ratio). I.e., PBL NOx MASS = NOX EMISSIONS * NOX PHOTOCHEMICAL LIFETIME. I understand that any error in wind direction results in the inference of a biased NOx lifetime value, but that doesn't mean that the gradient has no bearing on the rate at which NOx is being removed from the boundary layer. The question

instead should be "what factors affect the spatial pattern of NO2 column downwind of a city that should be accounted for when estimating the NOx photochemical lifetime.

The authors should also note here that there are oscillating thermally-driven wind flows in the basin that are challenging to resolve and may also lead to biased lifetime calculation. In the current version of the manuscript, these effects are only discussed in the conclusions section (P15 L16-25 of annotated revised manuscript). I would also disagree with the authors: there would be systematic impacts of local circulations on the spatial pattern of NO2 columns that would not necessarily be captured in the larger scale synoptic flow patterns. During night, air would flow downslope and out over the Yellow Sea. During daytime, the flow would reverse to the E. In general, the large scale flows during this time of year would be northwesterly. I don't expect the authors to fully account for these but to acknowledge that they may exist.

This paragraph in Section 3.6.2 has now been revised. We have also removed the statement in the discussion suggesting that there is no consistent bias in the top-down $NO_X$ emissions calculation.

We have now more explicitly stated that heterogeneous topography and oscillating thermally driven wind flows are effects that may bias the effective photochemical lifetime calculation.  We partially accounted for this bias by only selecting days with strong synoptic winds; on days with faster winds speeds, the sea & mountain breeze effects are secondary to the synoptic flow.  The last two sentences of the Section 3.6.2 paragraph have been removed. Please see the updated paragraph below:

"It should be noted that the NO2 photochemical lifetime derived here is a fundamentally different quantity than the NO2 lifetime observed by in situ measurements (de Foy et al., 2014; Lu et al., 2015) or derived by model simulations (Lamsal et al., 2010).  This is because the lifetime calculation is extremely sensitive to the accuracy of the wind direction (de Foy et al., 2014) and spatial pattern of the emissions. Inaccuracies in the wind fields introduce noise that shorten the tail of the fit.  As a result, NO2 photochemical lifetimes derived here are considered "effective" photochemical lifetimes and are generally shorter than the tropospheric column NO2 lifetimes derived by model simulations (Lamsal et al., 2010).  NOx sources at the outer portions of urban areas will lead to an artificially longer NO2 lifetime. This partially compensates for the bias introduced by the wind direction.   The heterogeneous topography and oscillating thermally driven wind flows (such as the Yellow Sea breeze) in the Seoul metropolitan area are effects that may bias the effective photochemical lifetime calculation.  We partially account for this bias by only selecting days with strong winds (>3 m/s); on days with faster winds speeds, the sea & mountain breeze effects are secondary to the synoptic flow."

P13 L19 – L30: "It should be noted that the NO2 photochemical lifetime derived here is a fundamentally different quantity than the NO2 lifetime observed by in situ measurements (de Foy et al., 2014; Lu et al., 2015) or derived by model simulations 20 (Lamsal et al., 2010). This is because the lifetime calculation is extremely sensitive to the accuracy of the wind direction (de Foy et al., 2014). Inaccuracies in the wind fields introduce noise that shorten the tail of the fit. As a result, NO2 photochemical lifetimes derived here are considered "effective" photochemical lifetimes and are universally shorter than the tropospheric column NO2 lifetimes derived by model simulations (Lamsal et al., 2010). NOx sources at the outer portions of urban areas will lead to an artificially longer NO2 lifetime. This partially 25 compensates for the bias introduced by the wind direction. The effective photochemical lifetime is also different from the NO2 lifetime derived by in situ measurements of NO2 at the surface or within the

boundary layer. In the boundary layer, NO2 is consumed faster yielding lifetimes that are shorter than the lifetimes based on tropospheric columns (Nunnermacker et al., 2007)."

**Review #2**

The revised manuscript by Goldberg et al. does an admirable job of addressing reviewer questions and following up on most reviewer suggestions. This includes additional simulations (longer simulations, additional emissions perturbation experiment) and evaluations (e.g., NOy), so the changes go beyond just text revisions, for which the authors are commended. Overall the paper is much improved, and I recommend it for publication in ACP, following some revisions to the text to address two points that could use further clarification, described below. Addressing these will not likely warrant further peer review.

Thank you for your comments.

Comments:

I appreciate the additional detail added in section 2.1.1 regarding how WRF-Chem profile biases are corrected. The details state that in each grid box, this bias is corrected via comparison to in situ aircraft observations. Are there unique profiles from aircraft measurements in each grid box though? I don't think so, as that would have required a lot of aircraft spirals. So, how is it determined which aircraft profile to use to correct the model values in each grid box? If comparing to the profiles measured by the aircraft throughout the entire "Seoul" or "mainland" transects (ie shown in Fig 5), how representative are these of model profiles in individual grid boxes? Some additional explanation and discussion of these issues is warranted.

We use a campaign mean comparison over all land-based areas and scale all modeled profiles by this ratio. As you correctly identified, there are not enough measurements in any one grid box to scale each individual model grid cell differently. Without more measurements, we are unsure how using a campaign-long mean comparison instead of a daily or location-specific comparison will affect the AMF calculation. This is beyond the scope of this paper. The revised sentence is below:

"We use a campaign mean comparison over all land-based areas (34° – 38° N, 126° – 130° E) and scale all modeled profiles in this box by this ratio; there are not enough measurements in any one grid box to scale each individual model grid cell differently."

Despite an attempt at clarifying in the revisions and response, I still don't follow what the authors are trying to demonstrate with discussion of NO2 emissions diurnal profiles and Figure 4. They are not clear regarding what they are suggesting. Do they think the diurnal variability in the model is incorrect? It seems though this is not an issue, as they show the emission rate at the model overpass time is representative of the 24 hr average. Do they suspect this is fortuitous? I'm not sure what they are suggesting as to why this agreement would explain a model underestimate of NO2 columns. Further, the inclusion of a plot of diurnal variability of NOx emissions in the US has yet to be strongly connected to whatever argument is being attempted here. Diurnal profiles of NOx emissions are different in different parts of the world — I'm not sure including a figure adds much to that statement. Thus, I still contend that Fig 4 and associated discussion be removed. Lastly, "temporalization" is not a word — suggest "temporal allocation" or "diurnal variability." And on this topic, the authors are mistaken in their response that they are the first to consider that model errors in diurnal variability could bias inversions

based on LEO satellites; rather, this is a common concern and a justification for the upcoming GEO satellites.

The temporal allocation of bottom-up emissions inventories can be a very significant source of uncertainty when using LEO satellites that only observe the mid-afternoon emissions rate. As you correctly identified, this was always a concern in the $NO_X$ remote sensing community that uses LEO satellites, but we have now put a loose bounds on this uncertainty. We are showing that there are scenarios in which the temporal allocation can be up to 35% different! The magnitude of this difference was much larger than we thought. We are not suggesting that the Korean emissions inventory should have the diurnal profile of the US or vice versa, but instead that there are scenarios in which the temporal allocation can vary widely. This further justifies the use of GEO satellites, perhaps even more than before.

We have modified the discussion in Section 3.3 to better clarify our thoughts and changed the word "temporalization" to temporal allocation. Please see below:

"Third, the temporal allocation of bottom-up emissions inventories can be a very significant source of uncertainty (Mues et al., 2014). The temporal allocation of the bottom-up Korean NOX emissions is such that the early afternoon rate during the OMI overpass time (between 12:00 – 14:00 local time) is approximately equal to 24-hour averaged rate (Figure 4). For comparison, in the eastern US, the early afternoon emission rate is 1.35 larger than the 24-hour averaged emission rate. Thus, there are scenarios in which the temporal allocation can be up to 35% different in the mid-afternoon during the OMI overpass time. We are not suggesting that the Korean emissions inventory should have the diurnal profile of the US or vice versa, but instead that there are scenarios in which the temporal allocation can vary widely. This substantiates the future use of geostationary satellites to better constrain this temporal allocation uncertainty. Lastly, the remaining difference will likely be due to an underestimate in the emissions inventory."

[revised manuscript text omitted]
 bottom-up emissions inventories can be a very significant source of uncertainty (Mues et al., 2014). The temporal allocation of the bottom-up Korean $NO_X$ emissions in this WRF-Chem simulation is such that the early afternoon rate during the OMI overpass time (between 12:00 – 14:00 local time) is approximately equal to 24-hour averaged rate (Figure 4). For comparison, using SMOKE in the eastern US yields an, the early afternoon emission rate that is 1.35 larger than the 24-hour averaged emission rate. Thus, there are scenarios in which the temporal allocation can be up to 35% different in the mid-afternoon during the OMI overpass time. We are not suggesting that the Korean emissions inventory should have the diurnal profile of the US or vice versa, but instead that there are scenarios in which the temporal allocation can vary widely. This substantiates the future use of geostationary satellites to better constrain this temporal allocation uncertainty. 
[revised manuscript text omitted]

30    topography and oscillating thermally driven wind flows (such as the Yellow Sea breeze) in the Seoul metropolitan area are effects that may bias the effective photochemical lifetime calculation. We partially account for this bias by only selecting days with strong winds (>3 m/s); on days with faster winds speeds, the sea & mountain breeze effects are secondary to the synoptic flow.

**3.7. Model simulation with increased NO$_X$ emissions**

5    To test whether an increase in the NO$_X$ emission rate is appropriate for the Seoul metropolitan area, we conduct a simulation with NO$_X$ emissions in the Seoul metropolitan area – within a 40 km radius of the city center – increased by a factor of 2.13, and analyze the results for May 2016. The 2.13 increase is representative of the change suggested by the top-down method (OMI-Regional: 484 kton/yr vs. WRF-Chem original: 227 kton/yr). This simulation was performed slightly differently than the original simulation in that it was a continuous month-long simulation and the outer domain was nudged to the reanalysis.

10    When comparing the new model simulation to in situ observations from the UC-Berkeley NO$_2$ and NCAR NO$_y$ instruments aboard the DC-8 aircraft, we find that NO$_2$ concentrations are a bit high, but NO$_y$ concentrations are in good agreement with WRF-Chem in the boundary layer when spatially and temporally collocated in the immediate Seoul metropolitan area (Figure 9). The NO$_2$-NO$_y$ partitioning is captured well by both model simulations, and there is no significant change in the NO$_2$-NO$_y$ ratio when using increased NO$_X$ emissions.

[revised manuscript text omitted]

Mues, A., Kuenen, J., Hendriks, C., Manders, A., Segers, A., Scholz, Y., Hueglin, C., Builtjes, P., and Schaap, M.: Sensitivity of air pollution simulations with LOTOS-EUROS to the temporal distribution of anthropogenic 55    emissions, 
[revised manuscript text omitted]
 the $NO_2$-$NO_y$ ratio in the Seoul plume when coincident $NO_2$ and $NO_y$ measurements are available.

[Figure]

**Figure 10.** Same as Figure 3, but now showing the WRF-Chem simulation with $NO_x$ emissions in the Seoul metropolitan area increased by a factor of 2.13 in panel (b).

[Figure]

**Figure 11.** (a) The OMI-Standard product during the month of May 2016, (b) the OMI-Regional NO$_2$ product with the WRF-Chem air mass factor adjustment and spatial kernel during the same period, (c) same as (b) but using WRF-Chem NO$_2$ profiles scaled based on the aircraft comparison, and (d) same as (b) but using the WRF-Chem simulation with NO$_x$ in the Seoul metropolitan area emissions increased by a factor of 2.13.